# Latent Sketchpad: Sketching Visual Thoughts to Elicit Multimodal Reasoning in MLLMs

## Abstract

While Multimodal Large Language Models (MLLMs) excel at visual understanding, they often struggle in complex scenarios that require visual planning and imagination. Inspired by how humans use sketching as a form of visual thinking to develop and communicate ideas, we introduce **Latent Sketchpad**, a framework that equips MLLMs with an internal *visual scratchpad*. The internal visual representations of MLLMs have traditionally been confined to perceptual understanding. We repurpose them to support generative visual thought without compromising reasoning ability. Building on frontier MLLMs, our approach integrates visual generation directly into their native autoregressive reasoning process. It allows the model to interleave textual reasoning with the generation of visual latents. These latents guide the internal thought process and can be translated into sketch images for interpretability. To realize this, we introduce two components: a Context-Aware Vision Head autoregressively produces visual representations, and a pretrained Sketch Decoder renders these into human-interpretable images. We evaluate the framework on our new dataset MazePlanning. Experiments across various MLLMs show that Latent Sketchpad delivers comparable or even superior reasoning performance to their backbone. It further generalizes across distinct frontier MLLMs, including Gemma3 and Qwen2.5-VL. By extending model's textual reasoning to visual thinking, our framework opens new opportunities for richer human–computer interaction and broader applications.

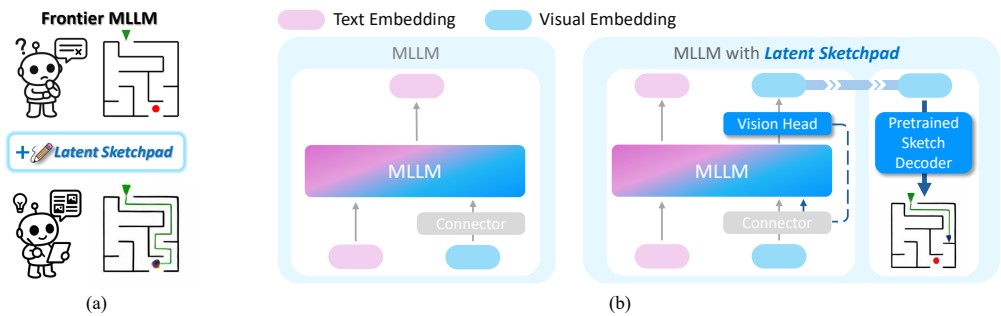

Figure 1: (a) Latent Sketchpad extends frontier MLLMs (e.g. Gemma3 and Qwen2.5-VL) to interleave text and visual latents generation, incorporating visual thoughts into reasoning. (b) The framework enables interleaved generation by equipping the pretrained MLLM with a Vision Head to generate visual latents autoregressively. A separately pretrained Sketch Decoder visualizes these latents into interpretable sketch images.

## 1 Introduction

Multimodal Large Language Models (MLLMs) extend pretrained LLMs with sophisticated vision encoders (Bai et al., 2025; Team et al., 2025), demonstrating remarkable success on a wide range of understanding tasks (e.g. VQA) (Zhang et al., 2025b; Fu et al., 2024). Furthermore, reasoning

techniques such as Chain-of-Thought (CoT) (Wei et al., 2022) have enabled models to tackle complex challenges by generating step-by-step textual reasoning traces (Li et al., 2025c). However, current MLLMs still face difficulties when encountering more advanced multimodal reasoning scenarios, especially those requiring precise spatial reasoning and dynamic visual grounding (Zhang et al., 2025a; Yang et al., 2025a; Li et al., 2024).

Humans naturally overcome such challenges by leveraging internal visual sketches alongside language, using mental imagery to simulate scenarios, test alternatives, and refine plans (Bruyer & Scailquin, 1998; Pearson, 2002). This interplay between verbal and visual thinking is crucial for effective reasoning, as visual imagination provides complementary structure and clarity that language alone fails to convey (Paivio, 1991). Motivated by this, recent research has explored equipping MLLMs with visual thinking to enhance reasoning (Su et al., 2025c).

One common strategy for enhancing multimodal reasoning is to interface with external visual tools, such as object detectors (Zheng et al., 2025; Su et al., 2025a) or executable code generators (Hu et al., 2024b; Wu et al., 2025a). However, these approaches are constrained by predefined tool capabilities and dependence on external environments. Recent efforts such as MVoT (Li et al., 2025a) have explored synthesizing intermediate visual outputs to aid reasoning. To validate its effectiveness, MVoT employs unified generative architectures capable of producing both text and images. But these models (Deng et al., 2025; Tong et al., 2024; Chern et al., 2024) are fundamentally oriented toward pixel-level rendering. Their training objectives prioritize image realism over visual abstractions most conductive for reasoning. In parallel, frontier pretrained MLLMs like Qwen2.5-VL and Gemma3 (Bai et al., 2025; Team et al., 2025) excel at perceptual understanding through large-scale vision–language pretraining. However, they lack the native ability to generate visual content as part of their reasoning process. Critically, leveraging their pretrained visual features to actively produce visual thought for enhancing reasoning also remains largely unexplored. This gap prompts the question: *Can the pretrained visual features of powerful MLLMs be repurposed as a generative sketchpad to enable more complex multimodal reasoning?*

To address the limitations of existing approaches, we propose **Latent Sketchpad**, a simple yet effective framework that extends pretrained MLLMs to integrate visual thoughts into their reasoning process, as illustrated in Figure 1(a). Inspired by human mental sketching for complex reasoning, Latent Sketchpad enables the model to generate continuous visual latents within its reasoning trajectory. Rather than decoding into images, these latents remain in the latent representation space during reasoning. Furthermore, our approach seamlessly integrates visual reasoning into the MLLM's autoregressive generation loop, without compromising its multimodal understanding capabilities.

Specifically, as illustrated in Figure 1(b), we introduce a **Context-Aware Vision Head**, which is responsible for generating visual latents at each reasoning step. It is conditioned not only on the current hidden state but also on the previous visual representations. This design allows the model to maintain visual coherence and refine its internal visual representation based on both inter- and intra-image contextual cues. To make these visual representations human-interpretable, we further propose a standalone **Sketch Decoder**, pretrained to render visual latents into sketch-style images. This enables inspection of the model's evolving reasoning trajectory, offering interpretable insight into the model's internal visual thought process. Together, these components endow the MLLM with the ability to generate visual latents during reasoning and to render them into explicit, human-interpretable images. To evaluate the effectiveness of our framework, we construct a MAZEPLANNING dataset featuring complex, interleaved multimodal reasoning trajectories. Experimental results demonstrate that Latent Sketchpad preserves the reasoning strength of pretrained MLLMs while augmenting it with interpretable visual traces. Moreover, Latent Sketchpad exhibits broad applicability, enabling models such as Gemma3 and Qwen2.5-VL to reason beyond text through internal visual generation.

The main contributions of this paper include:

- We propose Latent Sketchpad, a framework that equips pretrained MLLMs with a Vision Head to interleave the autoregressive generation of visual latents and text, thereby enhancing their ability to perform complex multimodal reasoning beyond language-only deliberation.

- We introduce a pretrained Sketch Decoder that faithfully visualize the pretrained visual features into images for transparent inspection of internal reasoning steps, and is broadly compatible with diverse pretrained vision encoders like CLIP and SigLIP.

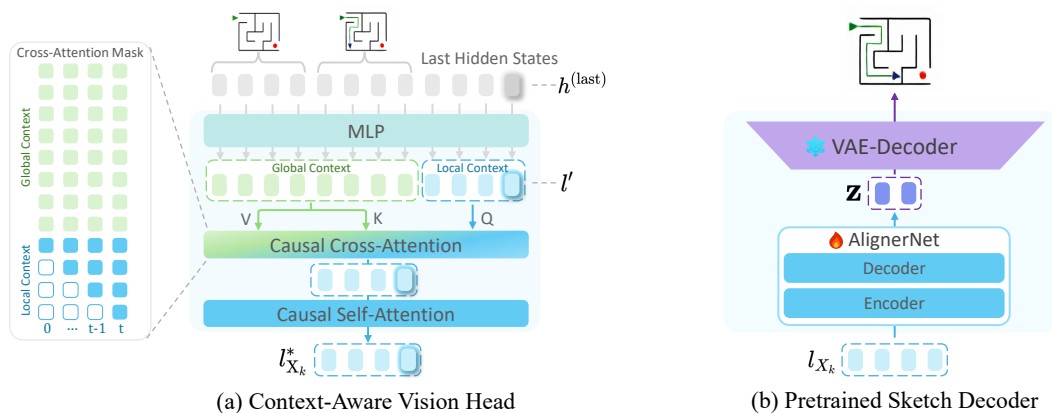

(a) Context-Aware Vision Head      (b) Pretrained Sketch Decoder

Figure 2: Architecture of the Context-Aware Vision Head and Sketch Decoder. The Vision Head transforms hidden states from the MLLM backbone into visual latents. The Sketch Decoder operates independently, converting these latents into sketch-style images for visualization and interpretability.

- We validate the effectiveness of Latent Sketchpad through comprehensive evaluations and analysis. The results show that our approach yields interpretable visual traces while retaining plug-and-play modularity and broad applicability across diverse pretrained MLLMs.

## 2 LATENT SKETCHPAD

To solve complex problems, humans often go beyond language, creating internal mental sketches to organize thoughts and visualize solutions. Inspired by this dual-modality process, we propose Latent Sketchpad, a framework that enables MLLMs to 'think' visually by repurposing pretrained visual features to generate continuous visual latents alongside text. By integrating linguistic and visual representations, Latent Sketchpad enhances reasoning with greater expressiveness and interpretability.

### 2.1 OVERVIEW

In the connector-based MLLM, a pretrained vision encoder encodes an input image $X_0$ into a sequence of latent visual tokens: $l_{X_0} = \mathcal{G}(X_0) \in \mathbb{R}^{n_v \times d_v}$, where $n_v$ denotes the number of visual tokens and $d_v$ is the dimensionality of each token. A connector module, as illustrated in Figure 1, projects these visual latents into the LLM's embedding space: $h_{X_0} = \mathcal{C}(l_{X_0}) \in \mathbb{R}^{n_v \times d_h}$, where $d_h$ denotes the dimensionality of LLM's embedding. The resulting visual embeddings $h_v$ are then concatenated with text embeddings $h_t$, forming a multimodal input sequence.

Our framework, as depicted in Figure 1, builds upon frontier MLLMs by introducing two new components:

- **Context-Aware Vision Head**: This vision head is integrated into the backbone. By leveraging previous visual features in the context, it generates context-aware visual latents from the internal hidden states of the backbone, reflecting the model's evolving mental images.
- **Pretrained Sketch Decoder**: The decoder operates independently of the MLLM and serves as a visualizer. By aligning the feature space of pretrained vision encoder with the latent space of pretrained VAE, it can translate the generated visual latents into sketch-style images.

With the Vision Head, the model can interleave textual and visual latent generation during the autoregressive generation of multimodal reasoning traces. Meanwhile, the Sketch Decoder serves as a visualization module, converting these internal latents into sketches. Together, our Latent Sketchpad supports interpretable and flexible multimodal reasoning.

### 2.2 CONTEXT-AWARE VISION HEAD

To interleave visual and textual reasoning within the the autoregressive generation, we introduce a Context-Aware Vision Head. While the hidden state of the MLLM backbone provides prior

context information, fine-grained visual details may become attenuated during long-range multimodal reasoning. To address this, the Vision Head explicitly perform visual generation by leveraging both:

1) *Global Context*: the latents of all preceding images, serving as long-range visual memory.
2) *Local Context*: the partial latents already produced within the current image, capturing short-term visual continuity.

Through the Vision Head, the resulting context-enriched visual latents can be projected into the language embedding space for continued autoregressive generation. Besides, they can also be decoded by our pretrained Sketch Decoder to produce interpretable sketch images.

**Auto-regressive Visual Latent Generation.** The visual generation process begins with a special `<start_of_image>` token, indicating the start of a new image. Following this signal, the model enters an auto-regressive loop to generate the visual latents $l_{X_k}$ for image $X_k$, one token at a time. When generating the $t$th image token, as illustrated in Figure 2 (a), the Vision Head first collects hidden states from global context $\{h_{X_j}^{(\text{last})}\}_{j=0}^{k-1}$ and local context $\{h_{X_k,i}^{(\text{last})}\}_{i=1}^{t}$. Then all these hidden states are projected into visual latent space as $\{l'_{X_j}\}_{j=0}^{k-1}$ and $\{l'_{X_k,i}\}_{i=0}^{t}$, respectively.

Let $L_{X_k}^{\text{global}} = [l'_{X_0}, l'_{X_1}, \ldots, l'_{X_{k-1}}]$ denote the global context latents, and $L_{X_k,t}^{\text{local}} = [l'_{X_k,0:t-1}, l'_{X_k,t}]$ represent the local context latents. Here $l'_{X_k,0:t-1}$ are the visual latents from previous steps within $X_k$ and $l'_{X_k,t}$ is the current latent at $t$. To incorporate contextual knowledge into current latent generation, the Vision Head performs causal cross-attention on $L_{X_k}^{\text{local}}$ and $L_{X_k}^{\text{global}}$, as illustrated in Figure 2 (a). Specifically, each token in the local context attends only to tokens preceding it across the entire sequence, thereby retrieving relevant visual cues from previously generated segments. This causal structure ensures that visual latents are generated in an autoregressive manner, with each image token conditioned on prior context. Subsequently, a causal self-attention is applied over the current image's local context latents $L_{X_k}^{\text{local}}$, ensuring coherence within the current image.

The resulting context-enriched latent, $l_{X_k,t}^*$, is then projected back into the language embedding space to auto-regressively predict the next token. This process iterates until a fixed number $n_v$ of visual tokens are generated, forming the complete latent sequence $l_{X_k}^* = \{l_{X_k,i}^*\}_{i=0}^{n_v-1}$. The visual generation concludes with the `<end_of_image>` token, after which text generation continues.

**Loss.** To supervise the Vision Head, we apply a latent-level regression loss between the predicted context-enriched latent $l_{X_k}^*$ and the target latent $l_{X_k}$. The target latent $l_{X_k}$ is extracted from the image in the intermediate visual thought using the pretrained vision encoder of MLLMs, providing a ground-truth latent representation for supervision. The loss can be instantiated using various similarity or distance measures (e.g., cosine similarity or L1 distance):

$$\mathcal{L}_{\text{reg}} = \mathcal{D}(l_{X_k}^*, l_{X_k}), \tag{1}$$

where $\mathcal{D}(\cdot, \cdot)$ denotes a generic latent regression criterion.

**Training.** The Vision Head is trained from scratch using the regression loss $\mathcal{L}_{\text{reg}}$, while keeping all parameters of the MLLM frozen. This training scheme isolates the learning of visual latent generation from the backbone, thereby preserving the original reasoning capacity of the MLLM.

## 2.3 PRETRAINED SKETCH DECODER

To support transparent and interpretable multimodal reasoning, we introduce a pretrained Sketch Decoder that converts pretrained visual features into human-interpretable sketches.

**Latent-to-Pixel Projection.** The Sketch Decoder is designed as a standalone visualization module, capable of decoding visual features obtained from pretrained ViT based vision encoder. As illustrated in Figure 2(b), the core component of the Sketch Decoder is a learnable alignment network (Pan et al., 2024), which is implemented as a Transformer-based architecture comprising an encoder and a decoder. It projects the visual latents into the latent space of a pretrained VAE. Specifically, since ViT features and VAE latent representations reside in distinct semantic spaces, the AlignerNet serves as a mapping function, transforming the visual tokens into latent vectors. For example, a sequence of visual latents $l_{X_k}$ is projected by the AlignerNet into VAE-compliant latent codes $\hat{z}$. These transformed codes are subsequently fed into a frozen VAE decoder to generate the corresponding pixel-space image $X_k$.

**Loss.** Given a training image $\mathbf{x}$ and its foreground mask $\mathbf{m} \in \{0,1\}^{H \times W}$, we first obtain target latent posterior $q(\mathbf{z} \mid \mathbf{x})$ from the frozen VAE encoder $E_{\text{VAE}}$. Meanwhile, the vision encoder extracts visual tokens, which are processed by AlignerNet to predict the parameters $(\boldsymbol{\mu}, \boldsymbol{\sigma})$ of a Gaussian distribution $q'(\hat{\mathbf{z}}) = \mathcal{N}(\boldsymbol{\mu}, \boldsymbol{\sigma}^2)$. The latent $\hat{\mathbf{z}} \sim q'$ is then decoded by the frozen VAE decoder $D_{\text{VAE}}$ to produce a reconstruction $\hat{\mathbf{x}} = D_{\text{VAE}}(\hat{\mathbf{z}})$. Together, these losses ensure alignment at both pixel and latent levels:

$$\mathcal{L} = \mathcal{L}_{\text{rec}} + \mathcal{L}_{\text{latent}} + \mathcal{L}_{\text{emb}}, \tag{2}$$

where: $\mathcal{L}_{\text{rec}} = \text{Focal}(\hat{\mathbf{x}}, \mathbf{x}, \mathbf{m})$ is a focal reconstruction loss designed to put extra emphasis on foreground pixels where $m_{ij} = 1$; $\mathcal{L}_{\text{latent}} = \text{NLL}_{\mathcal{N}}(\boldsymbol{\mu}, \boldsymbol{\sigma}; \mathbf{z})$ is the negative log-likelihood loss (Tschannen et al., 2024) that encourages the predicted latent distribution to approximate the ground-truth posterior; $\mathcal{L}_{\text{emb}} = \frac{1}{N} \sum_{i=1}^{N} \|\mathbf{e}_i - \hat{\mathbf{e}}_i\|_2^2$ is a mse loss between predicted and target patch embeddings.

**Training.** We employ the decoder of SDXL-VAE (Podell et al., 2023) and use its encoder to provide target latent posterior during training. The transformer-based sketch decoder is trained from scratch, with both vision encoder and VAE model frozen. During pretraining, we use the Quick, Draw! dataset (Jongejan et al., 2016), which comprises 50 million sketch-style images across 345 categories.

# 3 EXPERIMENTS

## 3.1 EXPERIMENTAL SETUPS

**Data.** To evaluate complex multimodal reasoning capabilities, we construct a MAZEPLANNING dataset. It comprises 47.8K mazes of size from 3×5 to 5×5 for training, each accompanied by interleaved text-and-image reasoning sequences. Additionally, we provide a test set of 500 mazes within the same size range, further divided into an easy set (< 4×5) and a hard set (4×5 and 5×5) based on their size. Detailed dataset statistics and construction procedures are provided in the Appendix A.

**Models.** We employ Gemma3-12B and Qwen2.5-VL-7B as our backbone, enhanced with Latent Sketchpad and fine-tuned on MAZEPLANNING to support interleaved text-image generation. Both models are evaluated under two reasoning modes: text-only CoT and multimodal CoT. To enable this, we adopt a unified fine-tuning scheme that equips a single model to operate in both modes. During training, all images except the initial input are randomly masked with a fixed probability (0.5), exposing the model to a mixture of purely textual reasoning steps and interleaved text–image sequences. This allows a single checkpoint to naturally support both text-only and multimodal reasoning at inference time. Visual generation is supported by our Context-Aware Vision Head, which is trained with the backbone frozen, making it plug-and-play without compromising the pretrained reasoning capacity of MLLMs. We also evaluate several proprietary models including GPT-4o, o1, o4-mini, and o3-pro (tool)[1]. Full implementation details are provided in the Appendix B.

**Evaluation Metrics.** We extract the model-predicted action sequences by pattern matching the content enclosed between the `<actions>` and `</actions>` tags. We employ two complementary evaluation metrics: (1) *Success Rate (SR)* measures the proportion of test cases in which the model generates a complete and correct action sequence. (2) *Progress Rate (PR)* quantifies the ratio of consecutively correct actions, reflecting how far the model progresses before making its first mistake.

## 3.2 EXPERIMENTAL RESULTS

We evaluate Latent Sketchpad on two representative MLLMs and provide the results together with proprietary models in Table 1. Each model equipped with Latent Sketchpad is compared against its own backbone under a consistent training protocol, ensuring fair comparison. The complete results across diverse training configurations and maze sizes are provided in Appendix C.4.

**Proprietary models struggles with complex and dynamic multimodal reasoning tasks.** As shown in Table 1, the results show that even strong proprietary models (e.g., o4-mini, o3-pro) achieve less than 20% success rate on our MAZEPLANNING. In addition, their progress rates remain below 50%, underscoring the difficulty proprietary models face in complex and dynamic multimodal

---

[1] o3-pro (tool) refers to the version with access to external tools.

Table 1: Experimental results on MAZEPLANNING. *o3-pro (tool)* refers to the version with access to external tools. The Latent Sketchpad integrated with GPT-4o is trained with all Qwen2.5-VL weights frozen. The absolute improvement Δ of models equipped with Latent Sketchpad (*+LS*) are highlighted in blue . 🖼 and **T** denote text-only output and interleaved text-image output.

| Model | | Output | Success Rate(%) | | | Progress Rate(%) | | |
|---|---|---|---|---|---|---|---|---|
| | | | *Easy* | *Hard* | *Average* | *Easy* | *Hard* | *Average* |
| *Proprietary* | **o1** | **T** | 21.00 | 6.50 | 15.20 | 40.72 | 27.95 | 35.61 |
| | **o4-mini** | **T** | 28.33 | 6.50 | 19.60 | 49.88 | 32.61 | 42.97 |
| | **o3-pro (tool)** | **T** | 24.33 | 9.50 | 18.40 | 46.03 | 35.08 | 41.65 |
| | **GPT-4o** | **T** | 11.00 | 5.00 | 8.60 | 32.44 | 28.12 | 30.71 |
| | *+ LS (ours)* | 🖼 , **T** | *+5.67* | *+1.00* | *+3.80* | *+10.69* | *+6.61* | *+9.06* |
| *Fine-tuned* | **Gemma3** | **T** | 85.67 | 46.50 | 70.00 | 95.22 | 76.09 | 87.57 |
| | *+ LS (ours)* | 🖼 , **T** | *+2.67* | *+1.50* | *+2.20* | *+0.86* | *+0.05* | *+0.53* |
| | **Qwen2.5-VL** | **T** | 65.67 | 33.00 | 52.60 | 88.32 | 70.91 | 81.35 |
| | *+ LS (ours)* | 🖼 , **T** | *+0.33* | *+0.50* | *+0.40* | *+0.35* | *+0.44* | *+0.39* |

reasoning. These failures primarily stem from the model's inability to track evolving spatial states (detailed in Appendix C.2.1), underscoring the limitations of these models in complex reasoning tasks. Notably, when GPT-4o is equipped with our Latent Sketchpad, the generated visual traces provide complementary spatial cues that effectively guide its reasoning, yielding significant improvements in both success and progress rates. In particular, it achieves performance comparable to dedicated reasoning models and even surpasses o1 on progress rate.

**Latent Sketchpad demonstrates promising plug-and-play capability.** A key advantage of Latent Sketchpad lies in its modular architecture: the Vision Head can be trained independently and attached to MLLMs without altering their parameters. This preserves the backbone's original reasoning ability while seamlessly augmenting it with visual generation. Empirical results show that Latent Sketchpad can be attached to MLLMs without noticeable degradation in reasoning performance, while simultaneously enabling the generation of visual traces that support multimodal reasoning. Specifically, Latent Sketchpad yields measurable improvements on Gemma3 and preserves the already strong performance of Qwen2.5-VL, demonstrating its effectiveness without compromising reasoning ability. These results underscore Latent Sketchpad 's promising plug-and-play capability.

**Latent Sketchpad exhibits broad applicability across different MLLMs.** Our experiments demonstrate that Latent Sketchpad seamlessly adapts to diverse pretrained backbones, including Gemma3 and Qwen2.5-VL. Despite their architectural differences, Latent Sketchpad consistently enables these models to externalize internal visual features as explicit reasoning traces, thereby enhancing interpretability and extending their multimodal reasoning capacity. This highlights Latent Sketchpad as a generally applicable enhancement for diverse MLLMs.

## 4 DISCUSSION AND ANALYSIS

### 4.1 GENERALIZATION AND COMPATIBILITY OF THE PRETRAINED SKETCH DECODER

To assess the generalization ability of our pretrained Sketch Decoder, we evaluate its zero-shot reconstruction performance on unseen samples from the MAZEPLANNING test set. As shown in Figure 3 (a), the decoder achieves consistently high SSIM (Structural Similarity) scores across three representative vision encoders (OpenCLIP, Qwen2.5-VL, and Gemma3), demonstrating strong generalization. Notably, these encoders differ significantly in pretraining schemes: Qwen2.5-VL's encoder employs window attention and is trained from scratch, while Gemma3 adopts a SigLIP-initialized encoder, highlighting our Sketch Decoder 's compatibility with diverse ViT-based vision encoders. In addition, qualitative examples (Figure 3 (b)) further present the decoder's ability to reconstruct sketches with high structural fidelity. Additional examples are provided in Appendix C.5.

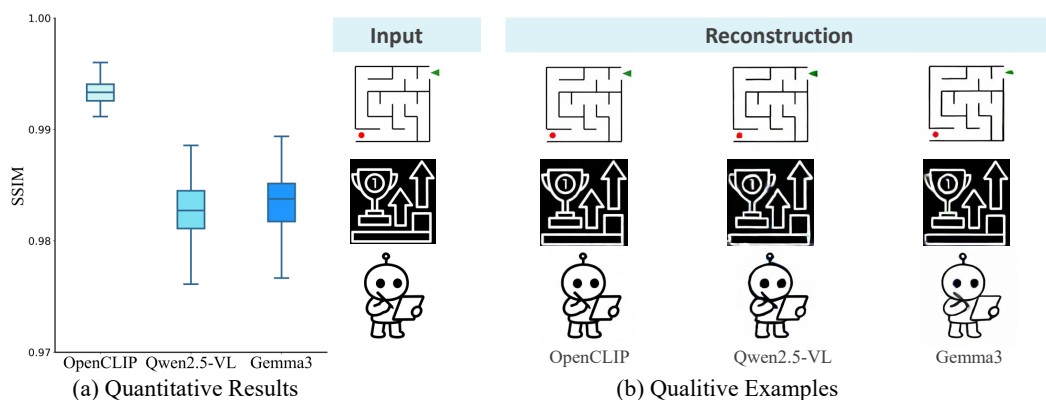

Figure 3: Illustration of generalization and compatibility of the pretrained Sketch Decoder. (a) Quantitative reconstruction results (SSIM) across different vision encoders (OpenCLIP, Qwen2.5-VL and Gemma3) on unseen samples from MAZEPLANNING. (b) Qualitative examples of reconstructed sketches from visual latents produced by each encoder.

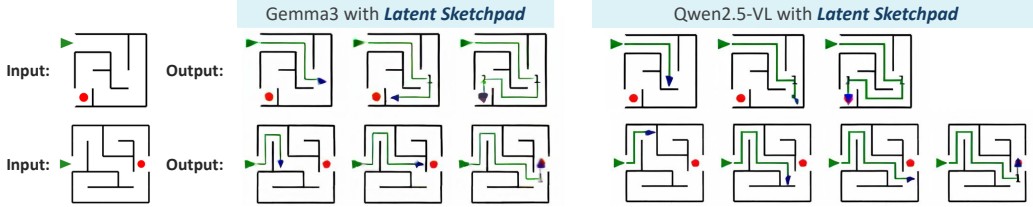

Figure 4: Qualitative analysis illustrating visualizations from Latent Sketchpad-enhanced Gemma3 and Qwen2.5-VL on in-distribution mazes. More examples are provided in Appendix C.6.

## 4.2 VISUALIZATION QUALITY IN DOWNSTREAM REASONING TASK

**Qualitative Analysis.** Figure 4 illustrates examples of visual thoughts generated by Latent Sketchpad-enhanced Gemma3 and Qwen2.5-VL on in-distribution test set. As shown in the figure, while the visualizations rendered via our Sketch Decoder may appear lower in perceptual quality, such as the arrows or digits, they exhibit great structural stability. This can be attributed to the Context-Aware Vision Head, which allows semantic context to dynamically guide the visual trajectory and enforce structural consistency throughout the planning process. More examples are provided in Appendix C.6.

**Quantitative Analysis.** To evaluate the quality of generated visual traces, we introduce two metrics:
- Layout Consistency Rate (LCR): whether the generated images preserve the spatial configuration of the maze, including the start point, end point, and wall placements
- Visual Success Rate (VSR): Assesses whether a valid path from the start to the goal is successfully drawn within the correct maze layout.

As summarized in Table 2, our Latent Sketchpad consistently performs well across different MLLMs. We highlight two key findings from these results: *(1) Latent Sketchpad preserves visual contextual consistency.* Across both models, Latent Sketchpad achieves notably high LCR, reflecting its stronger ability to maintain spatial structure throughout reasoning steps. This contextual stability enables MLLMs to plan valid paths, as evidenced by the correlation between layout consistency and VSR. *(2) Latent Sketchpad shows potential to support reasoning through visual generation.* For Gemma3 equipped with Latent Sketchpad, the VSR reaches 75.6%, substantially higher than the baseline SR of 70%. Therefore, as illustrated in Table 1, its performance is enhanced

Table 2: Quantitative results of visualization quality on MAZEPLANNING. *First* and *Last* refer to the first and final visualizations within a complete reasoning sequence, respectively.

|  | Layout Consistency Rate (%) | | | Visual Success Rate (%) |
|---|---|---|---|---|
|  | *First* | *Last* | *Overall* |  |
| Gemma3+LS | 99.40 | 99.20 | 99.34 | 75.60 |
| Qwen2.5-VL+LS | 99.80 | 98.60 | 98.77 | 66.60 |

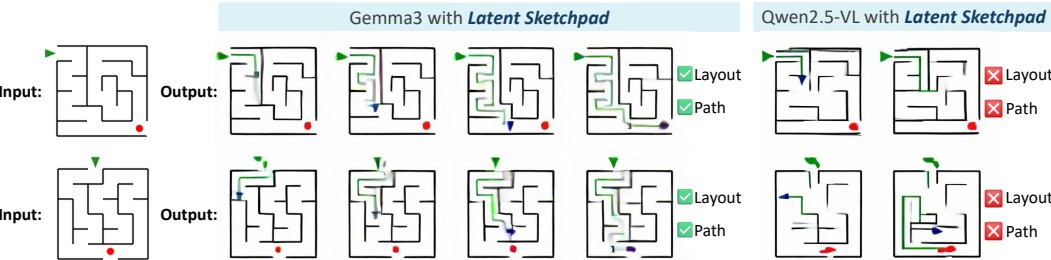

Figure 5: Visualizations from Latent Sketchpad on Gemma3 and Qwen2.5-VL in the OOD test set.

Table 3: Performance on the OOD test set of MAZEPLANNING.

|  | SR (%) | PR (%) |
|---|---|---|
| Qwen2.5-VL | 5.50 | 32.16 |
| Gemma3 | 8.00 | 38.76 |
| Gemma3+LS | **10.00** | **39.39** |

Table 4: Ablation results across different components.

|  | SR (%) | PR (%) | VSR (%) |
|---|---|---|---|
| Gemma3 *w/o adaptation* | 9.40 | 33.04 | - |
| Gemma3+LS | **72.20** | **88.10** | **75.60** |
| - *w/o augmentation* | 54.20 | 77.47 | 68.20 |
| - *w/ cosine* $\mathcal{L}_{\text{reg}}$ | 71.40 | 87.65 | 73.80 |

by the generated visual traces (70% to 72.2%). A consistent trend is also observed on Qwen2.5-VL, further confirming the ability of Latent Sketchpad to facilitate reasoning through visual generation.

### 4.3 FURTHER ANALYSIS

**Out-of-Distribution Generalization.** To further assess the generalization ability of Latent Sketchpad, we construct an OOD test set consisting of 200 mazes of size 6×6. Although fine-tuned Gemma3 and Qwen2.5-VL achieve strong performance on the in-distribution test set, their results drop sharply on the OOD set, as shown in Table 3. When equipped with Latent Sketchpad, Gemma3 shows improved robustness: it generates correct visual thoughts that yield performance gains (Table 3), with examples illustrated in Figure 5 and failure cases in Figure 11. However, Qwen2.5-VL fine-tuned with our limited data does not yet exhibit clear generalization with Latent Sketchpad. This is mainly due to Qwen2.5-VL constructs visual tokens by concatenating four encoded features before projection, in contrast to Gemma3, which pools them directly. This design produces a higher-dimensional input and demands substantially more data for generalization.

**Performance Across Maze Sizes**
As maze size increases, the evaluated models exhibit a notable decline in performance. As shown in Figure 6, this trend holds consistently across both proprietary models and Gemma3 equipped with our Latent Sketchpad. While our method maintains a higher success rate than the baselines across all maze scales, the increased spatial complexity in larger mazes presents a greater challenge for accurate planning.

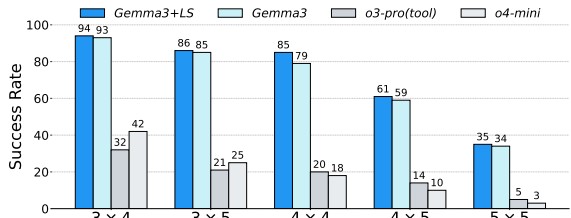

Figure 6: Performance Variation with Maze Size

### 4.4 ABLATIONS

We conduct a series of ablation studies to investigate the effects of modality alignment, data augmentation strategy, and different choices of regression loss on model performance.

**Effect of Connector Adaptation.** We investigate the impact of connector adaptation on model performance by analyzing whether the visual representations are updated during training. Taking Gemma3 as an example, freezing the connector severely impairs spatial understanding. The model often confuses directions such as left and right, leading to notable performance degradation as shown in the first row of Table 4. We also observe similar trends on Qwen2.5-VL. These findings highlight the critical role of connector adaptation during downstream task fine-tuning.

**Data Augmentation Improves Visual Accuracy and Task Performance.** To increase robustness, we introduce an augmentation strategy on the intermediate visual thoughts in the input of each training sample (detailed in Appendix B.4). The images are repeatedly reconstructed through our Sketch Decoder before being encoded, generating semantically equivalent but pixel-level perturbed views. This augmentation strategy preserves spatial semantics while injecting appearance variability, encouraging the model to focus on spatial structures. As shown in Table 4, the proposed augmentation improves the accuracy of visual thoughts and leads to higher task success rates.

**Choice of Regression Loss.** We compare L1 loss and cosine similarity as regression objectives for training the Vision Head. Empirically, we find that L1 loss consistently outperforms cosine similarity across all evaluation metrics. This suggests that directly minimizing element-wise distance in latent space better preserves the spatial and semantic fidelity in Latent Sketchpad.

## 5 RELATED WORK

**Multimodal Reasoning.** Recent studies have enhanced multimodal reasoning with visual inputs through Chain-of-Thought (CoT) prompting (Wei et al., 2022) or the use of external tools such as cropping and zooming (Zheng et al., 2025; Su et al., 2025b; Wu et al., 2025a; Fu et al., 2025), enabling more fine-grained visual perception during the reasoning process. These methods (Hu et al., 2024a; Fu et al., 2025) that invoke external tools to edit or manipulate images inevitably rely on predefined and limited action spaces, restricting their flexibility. Beyond tool-assisted approaches, methods like MVoT (Li et al., 2025a) and Visual Planning (Xu et al., 2025) generate visual thoughts natively for step-by-step reasoning, which demonstrate the feasibility and benefits of incorporating visual information as an additional modality for reasoning, complementing textual cues. While these methods reason across modalities in a generative manner, they typically rely on unified auto-regressive models trained for multimodal generation, often operating over discrete token sequences (Team, 2024; Chern et al., 2024). However, how to leverage the pretrained visual features of frontier MLLMs to generate visual thoughts remains largely underexplored. To address this gap, we propose Latent Sketchpad, a framework enabling pretrained MLLMs to generate visual latents, integrating visual thinking directly into its native autoregressive loop.

**Latent Reasoning.** Reasoning in large language models is often guided by explicit Chain-of-Thought (CoT) prompting, where verbalizing intermediate steps improves final accuracy (Wei et al., 2022). While effective, this approach is fundamentally constrained by the expressiveness of natural language. To overcome this, recent work on latent reasoning performs multi-step inference directly within the model's continuous hidden states, forgoing explicit token generation (Zhu et al., 2025). These methods, developed primarily for text, typically use architectural modifications for recurrent computation (Dehghani et al., 2018; Geiping et al., 2025) or training strategies that induce implicit reasoning steps (Hao et al., 2024; Tack et al., 2025). In multimodal scenarios, latent representation also helps to alleviate the modality gap by avoiding discretizing the image into visual tokens, with most previous work focusing on multimodal generation (Pan et al., 2025) instead of reasoning. Yang et al. (2025b) introduce latent visual tokens to enable multimodal reasoning, but their approach is still limited to generating one single image as the answer image during the reasoning process. In contrast, our Latent Sketchpad enables pretrained MLLMs to actively generate and utilize visual latents interleaved with textual rationales as internal reasoning steps.

**Unified Multimodal Generation.** Following recent advances in multimodal reasoning with textual outputs (Liu et al., 2024; Team et al., 2025; Bai et al., 2025), unified models capable of multimodal generation have begun to emerge (Wang et al., 2024; Chern et al., 2024; Wu et al., 2025b; Chen et al., 2025; An et al., 2025). These models extend output modalities beyond text to include images (Chern et al., 2024; Chen et al., 2025; An et al., 2024) and more (Zhan et al., 2024; Li et al., 2025b), typically through a combination of autoregressive modeling and diffusion-based image decoders. Rather than training a unified multimodal model from scratch, MetaMorph (Tong et al., 2024) introduces VPiT, which enables pretrained LLMs to understand visual inputs and generate a mixture of discrete text and continuous visual tokens. However, instead of reasoning, MetaMorph emphasizes image generation with surface-level semantics, which overlooks the intrinsic visual transitions within interleaved multimodal reasoning traces. In this work, we bridge that gap with a context-aware vision head by enabling an MLLM that already understands visual inputs to generate coherent multimodal reasoning traces without requiring extensive pretraining.

## 6 Conclusion

We introduce Latent Sketchpad, a simple yet effective framework that equips pretrained MLLMs with the ability to generate visual features as internal visual thoughts within their autoregressive reasoning loop. Inspired by the role of mental sketching in human cognition, Latent Sketchpad introduce a Context-Aware Vision Head to enable MLLMs to generate internal visual representations for enhanced reasoning, without relying on external tools. Additionally, a separately pretrained Sketch Decoder can be employed to translate these latent representations into interpretable sketches, facilitating human understanding and interaction. Extensive experiments show that Latent Sketchpad extends the reasoning capabilities of frontier MLLMs, enriching them with interpretable visual traces. Moreover, it shows broad applicability across diverse backbones, highlighting its potential as a general and plug-and-play enhancement. Our findings highlight the potential of integrating visual imagination directly into pretrained MLLMs, opening new avenues for more interpretable and capable multimodal systems.

## 7 Reproducibility Statement

We have made significant efforts to ensure the reproducibility of our work. The complete source code, including training and evaluation scripts, is provided in the supplementary materials in an anonymized form. Furthermore, detailed implementation specifications are presented in Appendix B, where we carefully describe model configurations, training procedures, and additional technical details. Together, these resources are intended to facilitate transparent verification and reproduction of our findings.

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

# A  MAZEPLANNING

To facilitate research on visual-language reasoning in complex environments, we construct a maze planning dataset that supports multimodal step-wise inference.

## A.1  DATASET OVERVIEW

The dataset comprises 47.8K unique mazes for training, each with varying grid sizes. For evaluation, we provide two distinct test sets: (1) an in-distribution (ID) test set of 500 mazes drawn from the same size distribution as the training data, and (2) an out-of-distribution (OOD) test set of 200 larger mazes with a fixed 6×6 grid configuration, designed to assess generalization to more complex scenarios.

Each maze instance is annotated with a multimodal trajectory that intertwines visual and textual reasoning steps. Unlike traditional grid-based formulations, we define action steps based on decision points to better reflect the natural, flexible reasoning process employed by humans. Specifically, we use the following three abstract action types:

- **Go forward**: Move straight until reaching the next decision point (e.g., an intersection or turn).
- **Turn left**: Rotate left before moving forward.
- **Turn right**: Rotate right before moving forward.

To enable dynamic visual grounding during inference—i.e., determining the agent's current location and verifying the correctness and plausibility of the inferred path—we segment the reasoning process into discrete states. Each state comprises a short sequence of $k \in [4, 6]$ actions, after which a rendered image of the agent's path so far is generated. The system then validates the inferred state: if the state is deemed valid and coherent, inference proceeds to the next state. To facilitate training, we decompose each maze's output label in the training set by individual states. During training, each sample is supervised to predict the reasoning process leading to the subsequent state. The complete statistics of our MAZEPLANNING dataset are provided in Table 1.

Table 5: Statistics of the MAZEPLANNING dataset.

| Grid Size | 3×4 | 3×5 | 4×4 | 4×5 | 5×5 | 6×6 |
|---|---|---|---|---|---|---|
| Action Length | 6.78 | 7.75 | 7.92 | 8.98 | 10.56 | 22.96 |
| State Length | 1.91 | 2.25 | 2.31 | 2.67 | 3.12 | - |
| Action Length of Each State | 5.04 | 5.12 | 5.16 | 5.24 | 5.42 | - |
| Train Set Size | 5,758 | 9,559 | 9,548 | 9,580 | 13,355 | 0 |
| Test Set Size | 100 | 100 | 100 | 100 | 100 | 200 |

## A.2  DATASET CURATION

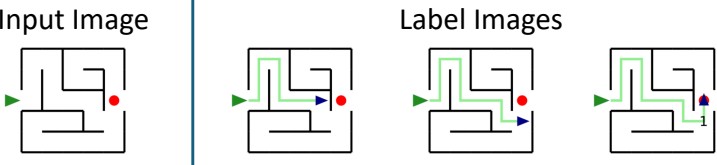

Figure 7: Input image and label images for the same sample in Table 6.

To ensure control over maze complexity and the interpretability of the reasoning process, we synthetically curated all maze samples and their corresponding annotations. Each maze was manually constructed to guarantee a single unique solution path from the start point to the goal. The layout of each maze was designed with varying grid sizes and branching patterns to create diverse decision-making scenarios, while maintaining the property of unambiguous solvability.

Based on the unique ground-truth trajectory of each maze, we manually annotated the sequence of actions (e.g. go forward, turn right and turn left) at key decision points. These annotations served as

Table 6: Textual reasoning steps for an example of MAZEPLANNING.

---

**MAZEPLANNING**

**Input Text:**
Given the maze in the input image <image>, determine a valid action sequence to navigate from the starting point (green arrow) to the endpoint (red circle). The black lines represent walls, and the white areas are traversable paths.
Each action in the sequence must be one of the following:
"go forward": Move straight until reaching the next turn or intersection.
"turn left": Rotate left before moving forward.
"turn right": Rotate right before moving forward.
  During the reasoning process, clearly mark each confirmed action using the format

<actions>confirmed action</actions>.
 **Label Text:**

Now, let's reason through the next 9 steps.
At the maze's starting point, a left turn corner presents itself, marking the initial curve in the path. Continuing along, a right turn corner is encountered, leading to another turn in the corridor. Subsequently, another right turn corner directs the path further along the maze. Finally, a left turn corner appears, guiding the way deeper into the labyrinth. Taking into account the visible layout of the maze, the next steps should be to move forward into the maze, then turn left and proceed forward, followed by a right turn and advance, another right turn and move forward, and finally a left turn to continue further into the maze.
The actions of this part are <actions>go forward, turn left, go forward, turn right, go forward, turn right, go forward, turn left, go forward</actions>
<image>
Let's continue.

Now, let's reason through the next 4 steps.
The path begins with a right turn corner, seamlessly transitioning into a new section of the maze. Continuing through this segment leads to a left turn corner, indicating another change in direction. Considering the structure of this maze section, the appropriate movement sequence is to first turn right and proceed forward, then make a left turn and continue moving forward, exploring deeper into the maze.
The actions of this part are <actions>turn right, go forward, turn left, go forward</actions>
<image>
Let's keep going.

Now, let's reason through the next 2 steps.
The path reaches the 1st junction, where the left path leads directly to the exit. Considering the structure of this maze section, the appropriate movement sequence is to turn left and proceed forward to reach the exit immediately.
The actions of this part are <actions>turn left, go forward</actions>
<image>
The inference process has concluded.

Table 7: Hyper-parameters of fine-tuning different models with various settings.

| Hyper-Parameters | Liquid$_T$ | Liquid | Gemma3 | LS of Gemma3 | Qwen2.5-VL | LS of Qwen2.5-VL |
|---|---|---|---|---|---|---|
| Random Seed | 42 | 42 | 42 | 42 | 42 | 42 |
| Epochs | 13 | 13 | 2 | 5 | 2 | 5 |
| Learning Rate | 0.0001 | 0.0001 | 0.0001 | 0.0001 | 0.0001 | 0.0005 |
| Global Batch Size | 128 | 128 | 128 | 128 | 128 | 128 |

Table 8: Model version of proprietary models.

| | GPT-4o | o1 | o4-mini | o3-pro |
|---|---|---|---|---|
| Model Version | 2024-11-20 | 2024-12-17 | 2025-04-16 | 2025-06-10 |

the foundation for generating the multimodal reasoning sequences. To simulate natural, human-like step-by-step reasoning, we employed GPT-4o to synthesize rich textual descriptions for each sample. Given the ground-truth action sequence, GPT-4o was prompted to produce coherent reasoning narratives that align with the intended visual path, effectively integrating spatial reasoning, language generation, and task context. The resulting data instances thus comprise tightly coupled image-text sequences, designed to reflect realistic and interpretable reasoning workflows. An illustrative example of this multimodal reasoning process is provided in Table 6 and Figure 7.

# B    IMPLEMENTATION DETAIL

## B.1    MODELS

The Context-Aware Vision Head consists of 2 layers of cross-attention, followed by 8 layers of self-attention. And the Sketch Decoder follows a standard encoder–decoder transformer architecture, which comprises 12 encoder layers and 12 decoder layers.

All the employed proprietary models are hosted on the Azure platform, with model version outlined in Table 7. We fine-tune both Qwen2.5-VL[2] and Gemma3[3] on our MAZEPLANNING dataset. Additionally, we also employ a discrete-token based unified MLLLM Liquid[4] for finetuning.

To support both text-only and multimodal chain-of-thought (CoT) reasoning within a unified framework, we design a fine-tuning scheme as follows. We fine-tune Gemma3-12B and Qwen2.5-VL-7B on a single source of reasoning trajectories from the MAZEPLANNING dataset, which contain interleaved text and image states. During training, all images except the initial input are randomly masked with a fixed probability (0.5). This strategy exposes the model to a mixture of purely textual reasoning steps and interleaved text–image sequences, allowing a *single* checkpoint to naturally operate in both text-only and multimodal modes at inference time. Visual generation is enabled through our Context-Aware Vision Head. This component is trained independently of the backbone. In this way, we preserve the original reasoning ability of the pretrained backbone while augmenting it with the capacity to generate visual thoughts.

During Inference, we do not modify the decoding process for text-only CoT. For multimodal CoT, however, we automatically insert a special token `<start_of_image>` during generation, which triggers the model to interleave textual and visual features. Specifically, on the MAZEPLANNING dataset, we append `<start_of_image>` immediately after each `</actions>` token, thereby enabling the model to generate the subsequent visual state.

## B.2    HYPER-PARAMETER

Table 7 shows the hyper-parameters for training Liquid, Qwen2.5-VL and Gemma3. All models were trained on MI300X GPUs. Table 7 provides the details of GPU configurations and hyperparameters

---

[2]https://huggingface.co/Qwen/Qwen2.5-VL-7B-Instruct
[3]https://huggingface.co/google/gemma-3-12b-it
[4]https://huggingface.co/Junfeng5/Liquid_V1_7B

Table 9: Example of prompt template.

---

**Prompt Template**

Given the maze in the input image <image>, determine a valid action sequence to navigate from the starting point (green arrow) to the endpoint (red circle). The black lines represent walls, and the white areas are traversable paths.

Each action in the sequence must be one of the following:
"go forward": Move straight until reaching the next turn or intersection.
"turn left": Rotate left before moving forward.
"turn right": Rotate right before moving forward.

During the reasoning process, clearly mark each confirmed action using the format <actions>confirmed action</actions>.

---

for various experimental settings. The backbone of Gemma3 and Qwen2.5-VL are both finetuned for 2 epoch. As detailed in Appendix C.4, we have explored different training setting for the connector. Furthermore, for the training of the Latent Sketchpad or the Sketch Decoder, all loss weights were set to 1.0.

All the employed proprietary models are hosted on the Azure platform, with model version outlined in Table 7.

### B.3 PROMPTING TEMPLATES

Table 9 shows an example of prompting templates and responses with different system variants.

### B.4 LATENT RECONSTRUCTION AUGMENTATION

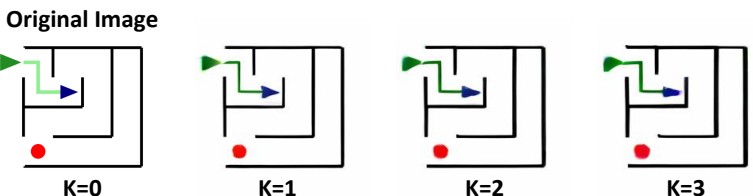

Figure 8: Step-wise reconstruction of the input image over $k$ iterations.

As described in Appendix A, the input of each training sample may include intermediate visual thoughts generated by the model. To improve the robustness of visual representations, we apply Latent Reconstruction Augmentation during training. Specifically, we repeatedly pass each input visual thought through the vision encoder and the pretrained decoder for up to $k$ rounds ($k \in [0, 3]$), reconstructing the image from its latent features in each step. This process preserves the semantic content while introducing minor perturbations in appearance, effectively encouraging the model to focus on stable spatial structures. The final reconstructed sketch is then used as the input image for training. Examples of this multi-step reconstruction are illustrated in Figure 8.

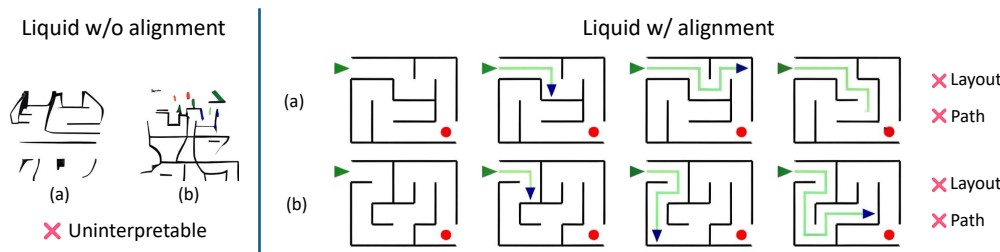

Figure 9: Failure cases from Liquid with and without modality alignment.

# C    ADDITIONAL EXPERIMENTS AND DISCUSSION

## C.1    EXPERIMENTS ON LIQUID

### C.1.1    IMPACT OF MODALITY ALIGNMENT

To investigate the impact of modality alignment during training, we compare two configurations of the Liquid model: one with the image embedding layer frozen (i.e., no alignment), and one with the image embedding layer unfrozen and trained jointly (i.e., with alignment). As shown in the left part of Figure 9, when the image embedding layer remains frozen, the model fails to establish meaningful correspondences between visual and textual modalities. This results in severely degraded outputs—uninterpretable images characterized by chaotic.

In contrast, enabling modality alignment by unfreezing the image embedding layer leads to a significant improvement in semantic coherence. As illustrated in the right part of Figure 9, the generated images become more structured and visually plausible, often matching the intended high-level semantics of the task.

### C.1.2    UNSTABLE LAYOUT CONSISTENCY

As illustrated in the right part of Figure 9, the model tends to alter the underlying maze layout itself. This structural inconsistency violates task constraints, as the generated solution no longer corresponds to the original maze, rendering the prediction incorrect despite its visual clarity. Such instability may stem either from the limited capacity of the base model or from the inherent brittleness of discrete token–based encoding.

## C.2    ERROR ANALYSIS

### C.2.1    PROPRIETARY MODELS

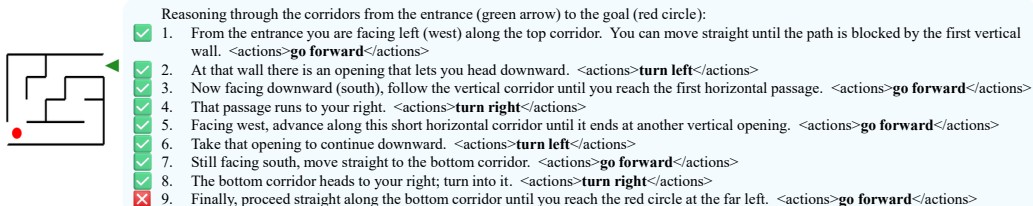

Figure 10: Failure cases of o3-pro (tool).

Despite the recent success of proprietary reasoning models in a wide range of complex tasks, their performance on our MAZEPLANNING benchmark reveals notable limitations, which is presented in Table 1. As illustrated in Figure 10, even o3-pro, a powerful reasoning model that supports external tool usage during inference, fails to solve certain maze navigation tasks. A key failure mode we observe is the model's inability to reliably localize itself during reasoning, especially in multi-step scenarios that require consistent visual tracking across states. Most models are able to correctly follow

Table 10: Task performance of the original Gemma3 and our fine-tuned Gemma3*.

| Model | Standard-Size Maze ($\leq 5\times5$) | | Extended-Size Maze ($6\times6$) | |
|---|---|---|---|---|
| | Success Rate (%) | Progress Rate (%) | Success Rate (%) | Progress Rate (%) |
| **Gemma3** | 5.80 | 24.15 | 0.50 | 11.76 |
| **Gemma3*** | 70.00 | 87.57 | 8.00 | 38.76 |

the initial steps. However, as the reasoning progresses and the agent moves deeper into the maze, these models often lose track of their spatial location, leading to compounding errors in path prediction and ultimately an incorrect final plan. These failures highlight a fundamental gap in current proprietary systems: while they excel at executing external tools and producing fluent responses, they often lack internal visual thought, a coherent internal representation of spatial progress and accumulated visual knowledge throughout a reasoning sequence. In contrast, our proposed Latent Sketchpad explicitly maintains and updates such an internal visual memory, enabling dynamic localization and more accurate path planning.

### C.2.2 LATENT SKETCHPAD

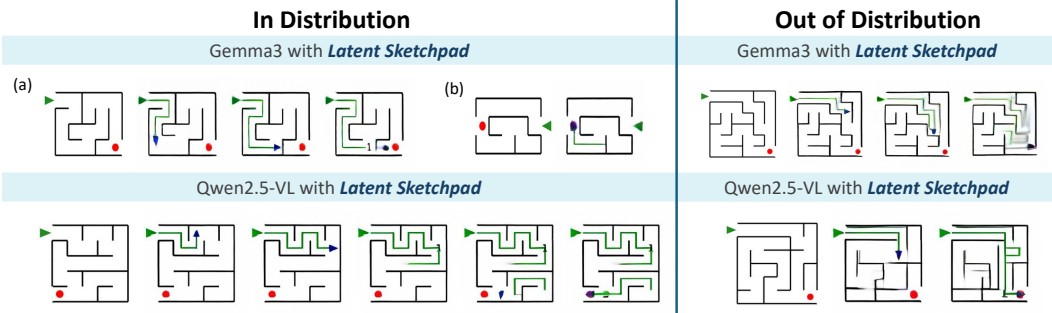

Figure 11: Failure cases of Latent Sketchpad.

To better understand the limitations of our proposed Latent Sketchpad framework, we conduct a qualitative error analysis under both in-distribution (ID) and out-of-distribution (OOD) settings.

In the ID setting, although the model performs well in most cases, we observe occasional failures where the predicted path exhibits spatial violations. As illustrated in the left part of Figure 11, the agent may generate trajectories that cut through maze walls or suddenly teleport to distant locations without following a physically valid path. These discontinuities often lead to incorrect final plans, despite the individual actions appearing locally coherent.

Furthermore, under the OOD setting (larger and unseen mazes), the model encounters a different failure mode. For Gemma3, this manifests as a gradual degradation of visual sketches, eventually causing the model to lose track of its position within the maze. In contrast, Qwen2.5-VL exhibits a different limitation: due to its vision encoder producing features four times larger than those of Gemma3, our limited fine-tuning data is insufficient to ensure generalization. As a result, Qwen2.5-VL fails to preserve maze layouts reliably and struggles to generate valid navigation paths.

These observations reveal two distinct types of failure: structural violations in familiar settings and cumulative degradation in novel environments, both of which point to potential avenues for future improvement in spatial consistency and robustness to distribution shifts.

### C.3 PERFORMANCE OF GEMMA3 ON MAZEPLANNING

As shown in Table 10, the base Gemma3 model exhibits limited performance on MAZEPLANNING, indicating insufficient capability for complex spatial reasoning. To address this, we first fine-tune the model using text-only data to build a foundational understanding. This step alone yields a substantial performance improvement, confirming the effectiveness of text-only supervision in enhancing baseline

Table 11: Success Rate of different system variants on MAZEPLANNING

| Grid Size | $3 \times 4$ | $3 \times 5$ | $4 \times 4$ | $4 \times 5$ | $5 \times 5$ | Overall |
|---|---|---|---|---|---|---|
| **GPT-4o** | 6.00 | 8.00 | 2.00 | 4.00 | 3.00 | 4.60 |
| **o1** | 31.00 | 16.00 | 16.00 | 11.00 | 2.00 | 15.20 |
| **o4-mini** | 42.00 | 25.00 | 18.00 | 10.00 | 3.00 | 19.60 |
| **o3-pro** | 32.00 | 21.00 | 20.00 | 14.00 | 5.00 | 18.40 |
| **Liquid$_T$** | 55.00 | 49.00 | 43.00 | 31.00 | 13.00 | 38.20 |
| **Liquid** | 91.00 | 72.00 | 75.00 | 52.00 | 29.00 | 63.80 |

Table 12: Progress Rate of different system variants on MAZEPLANNING

| Grid Size | $3 \times 4$ | $3 \times 5$ | $4 \times 4$ | $4 \times 5$ | $5 \times 5$ | Overall |
|---|---|---|---|---|---|---|
| **GPT-4o** | 23.75 | 22.50 | 21.39 | 20.10 | 16.14 | 20.78 |
| **o1** | 47.76 | 37.00 | 37.40 | 33.46 | 22.44 | 35.61 |
| **o4-mini** | 59.02 | 48.44 | 42.18 | 36.97 | 28.25 | 42.97 |
| **o3-pro** | 49.21 | 43.97 | 44.92 | 40.60 | 29.56 | 41.65 |
| **Liquid$_T$** | 74.51 | 70.32 | 68.25 | 60.98 | 43.63 | 63.54 |
| **Liquid** | 97.64 | 89.17 | 90.90 | 81.19 | 64.44 | 84.67 |

reasoning abilities. It also establishes a suitable backbone for directly equipping our Latent Sketchpad, enabling plug-and-play visual reasoning without requiring full model retraining.

We do not report results on Qwen2.5-VL in this setting, as its weaker instruction-following capability prevents us from obtaining consistent and meaningful outputs.

## C.4 TASK PERFORMANCE

To provide a comprehensive comparison across different model configurations, we report the task performance of all system variants on mazes of varying sizes. The results of proprietary models and Liquid are presented in Table 11 (success rate) and Table 12 (progress rate).

In addition, we conducted experiments under three connector tuning configurations for each model: (i) connector frozen throughout fine-tuning, (ii) connector unfrozen for one epoch, and (iii) connector unfrozen for two epochs. Our observations indicate that the two backbones exhibit distinct convergence behaviors, as illustrated in Table 13 and Table 14. When the connector remains frozen, both Qwen2.5-VL and Gemma-3 perform poorly. Allowing one epoch of connector tuning substantially improves Qwen2.5-VL, which adapts quickly, whereas Gemma3 still underperforms. In this regime, LS does not yield noticeable improvements on Gemma3 compared to Qwen2.5-VL, as the base model itself has not reached a sufficiently strong level of task performance.

When the connector is unfrozen for two epochs, Qwen2.5-VL achieves a strong performance, leaving limited headroom for further gains. In this case, adding Latent Sketchpad results in a visual success rate of 82.6, which is comparable to the text-only reasoning baseline (82.4) and thus brings little additional benefit. In contrast, Gemma3 benefits significantly from Latent Sketchpad under the same setting. With the visual success rate reaches 75.6, which is higher than its text-only baseline (70), the task performance of Latent Sketchpad enhanced Gemma3 increases to 72.2.

## C.5 ADDITIONAL QUALITATIVE EXAMPLES OF RECONSTRUCTION

As illustrated in Figure C.5, we present additional qualitative reconstruction results on unseen sketch-style samples. These examples span a variety of structural layouts and visual abstractions, and consistently demonstrate the decoder's ability to recover key geometric and semantic patterns from the visual latent space. While minor degradations in fine-grained line reconstruction and color fidelity are observed, the current performance is sufficient for supporting visual reasoning within the Latent Sketchpad. Future work may further enhance visual fidelity to expand applicability in tasks requiring finer perceptual precision.

Table 13: Success Rate of different system variants on MAZEPLANNING

|  | Connector | $3 \times 4$ | $3 \times 5$ | $4 \times 4$ | $4 \times 5$ | $5 \times 5$ | Overall |
|---|---|---|---|---|---|---|---|
| **Gemma3** | Frozen | 10.00 | 17.00 | 12.00 | 5.00 | 3.00 | 9.40 |
| **Gemma3** | 1 epoch | 52.00 | 30.00 | 21.00 | 23.00 | 8.00 | 26.80 |
| **Gemma3+LS** | 1 epoch | 51.00 | 30.00 | 24.00 | 21.00 | 7.00 | 26.60 |
| **Gemma3** | 2 epoch | 93.00 | 85.00 | 79.00 | 59.00 | 34.00 | 70.00 |
| **Gemma3+LS** | 2 epoch | 94.00 | 86.00 | 85.00 | 61.00 | 35.00 | 72.20 |
| **Qwen2.5-VL** | Frozen | 27.00 | 17.00 | 18.00 | 12.00 | 2.00 | 15.20 |
| **Qwen2.5-VL** | 1 epoch | 79.00 | 64.00 | 54.00 | 45.00 | 21.00 | 52.60 |
| **Qwen2.5-VL+LS** | 1 epoch | 79.00 | 63.00 | 56.00 | 45.00 | 22.00 | 53.00 |
| **Qwen2.5-VL** | 2 epoch | 98.00 | 96.00 | 95.00 | 79.00 | 44.00 | 82.40 |
| **Qwen2.5-VL+LS** | 2 epoch | 98.00 | 94.00 | 94.00 | 81.00 | 43.00 | 82.00 |

Table 14: Progress Rate of different system variants on MAZEPLANNING

|  | Connector | $3 \times 4$ | $3 \times 5$ | $4 \times 4$ | $4 \times 5$ | $5 \times 5$ | Overall |
|---|---|---|---|---|---|---|---|
| **Gemma3** | Frozen | 34.54 | 44.05 | 37.80 | 27.79 | 21.01 | 33.04 |
| **Gemma3** | 1 epoch | 65.94 | 55.84 | 51.11 | 48.40 | 33.64 | 50.98 |
| **Gemma3+LS** | 1 epoch | 65.11 | 54.56 | 51.84 | 47.93 | 32.68 | 50.42 |
| **Gemma3** | 2 epoch | 98.21 | 95.74 | 91.72 | 84.71 | 67.47 | 87.57 |
| **Gemma3+LS** | 2 epoch | 98.78 | 95.19 | 94.27 | 85.20 | 67.08 | 88.10 |
| **Qwen2.5-VL** | Frozen | 53.19 | 48.65 | 42.92 | 39.85 | 21.38 | 41.20 |
| **Qwen2.5-VL** | 1 epoch | 92.72 | 87.06 | 85.17 | 78.90 | 62.91 | 81.35 |
| **Qwen2.5-VL+LS** | 1 epoch | 92.92 | 86.93 | 86.16 | 78.22 | 64.47 | 81.74 |
| **Qwen2.5-VL** | 2 epoch | 99.46 | 98.26 | 98.61 | 93.44 | 77.98 | 93.55 |
| **Qwen2.5-VL+LS** | 2 epoch | 99.46 | 97.39 | 98.14 | 93.85 | 77.33 | 93.23 |

## C.6 VISUALIZATIONS

We additionally provide visualizations of the visual latents produced by the Latent Sketchpad on the MAZEPLANNING tasks, as presented in Figure 13. These examples, decoded via our pretrained Sketch Decoder, illustrate how the model leverages visual thoughts to organize spatial information and guide step-by-step decision making. The results demonstrate that even without photorealistic detail, the generated sketches capture sufficient structural cues to support accurate multimodal reasoning.

## C.7 DISCUSSION ON TRANSFERABILITY TO GENERAL TASKS

While our primary experiments focus on the MAZEPLANNING dataset, the proposed Latent Sketchpad framework is conceptually extensible to a wide range of multimodal reasoning tasks. MAZEPLAN-NING was chosen as an initial testbed because it provides both a challenging reasoning environment and controllable visual supervision, allowing us to rigorously validate the feasibility of native visual thought generation. However, the underlying mechanism is not tied to a specific domain. For general multimodal reasoning benchmarks such as MathVista or MMMU, our design philosophy emphasizes compatibility without compromise: the Latent Sketchpad can be attached to frontier VLMs to enable visual generation only when needed, while preserving their strong performance on conventional textual or perceptual reasoning tasks. In practice, this allows the model to handle simple reasoning through text alone and invoke the sketchpad for complex spatial or multi-step reasoning challenges. We view this adaptive integration as a promising direction for future research.

## D THE USE OF LARGE LANGUAGE MODELS

Large language models (LLMs) were used as general-purpose tools in this work. Specifically, LLMs assisted in (i) constructing reasoning trajectories for the MAZEPLANNING dataset and (ii) polishing the writing to improve clarity and readability.

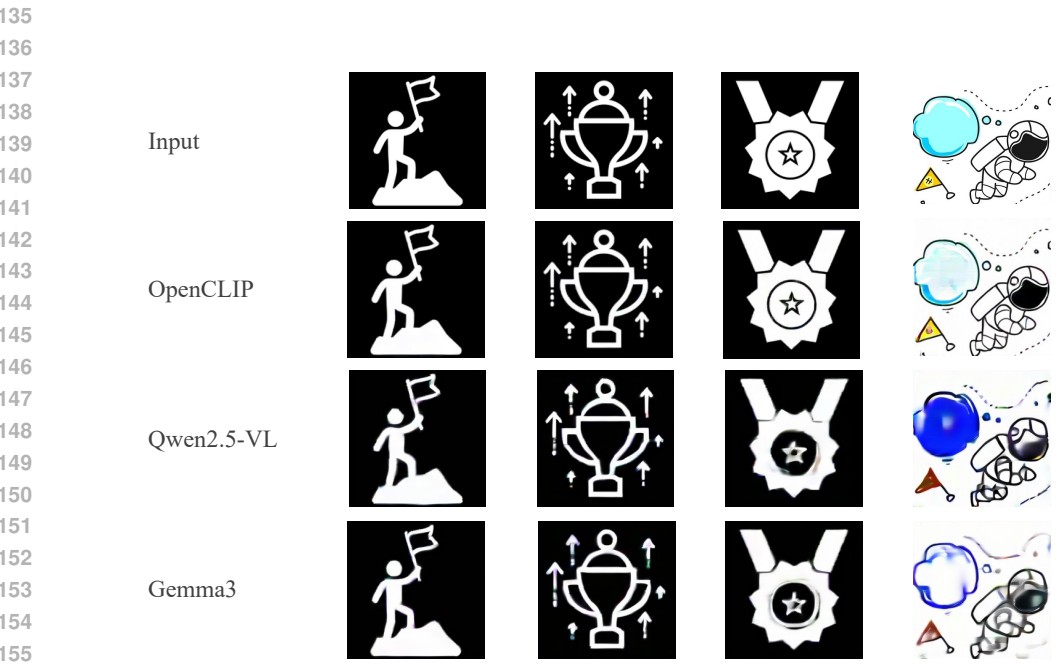

Figure 12: Additional qualitative examples of reconstructed sketches of Sketch Decoder.

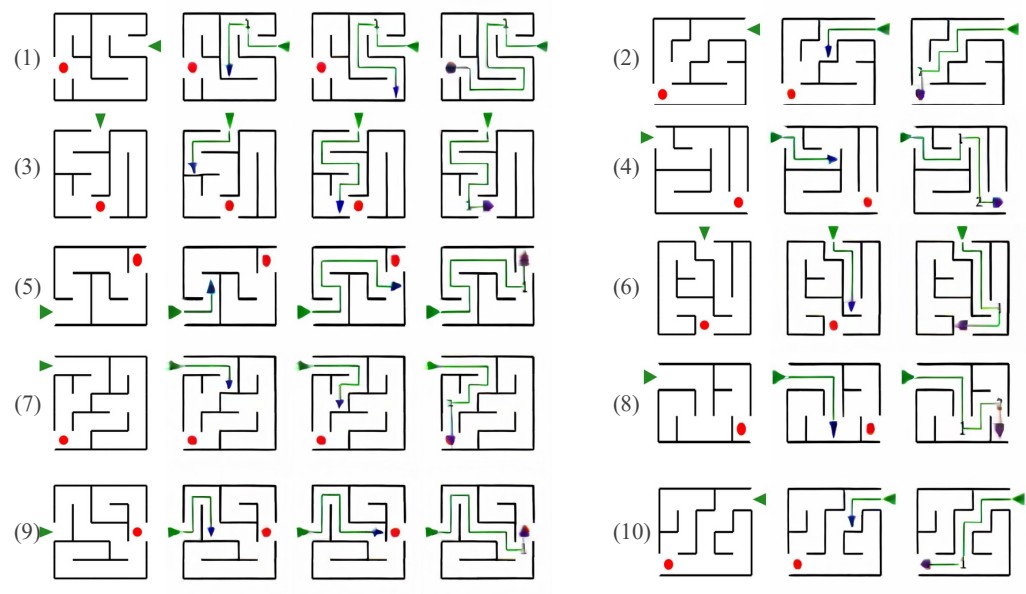

Figure 13: Examples of visual thoughts produced by Latent Sketchpad.

