# OpenReview forum: "Latent Sketchpad:  Sketching Visual Thoughts to Elicit Multimodal Reasoning in MLLMs"
_ICLR.cc/2026/Conference — Submitted to ICLR 2026_

### Official Review · Reviewer_QkdV · 2025-10-24

**Soundness:** 2
**Presentation:** 2
**Contribution:** 2
**Rating:** 6
**Confidence:** 4

**Summary:**

This paper introduces Latent Sketchpad, a modular framework designed to enable Multimodal Large Language Models (MLLMs) to perform visual reasoning by integrating visual latent generation into their native autoregressive process. The approach comprises a Context-Aware Vision Head for generating visual latents based on both local and global context, and a Sketch Decoder that converts these internal states into human-interpretable sketches. The framework augments existing MLLMs, such as Gemma3 and Qwen2.5-VL, without altering their core parameters. Evaluation on the newly constructed MAZEPLANNING dataset demonstrates that Latent Sketchpad preserves, and in some cases slightly improves, reasoning performance, while producing visual traces that support interpretability and facilitate multimodal reasoning. The method is validated through extensive analysis, ablations, and comparison to relevant baselines.

**Strengths:**

1. The paper addresses a significant gap in the current MLLM landscape by moving beyond pixel-level rendering and static perception, offering a pathway for internal visual thought processes within stepwise multimodal reasoning—an aspect not explicitly handled by existing MLLMs or unified autoregressive models.
2. Latent Sketchpad is implemented in a way that does not require end-to-end retraining or architectural changes to the main MLLM. This plug-and-play modularity, evidenced in experiments with both Gemma3 and Qwen2.5-VL, is practical for broad adoption.
3. The use of internal sketches, visualized via the Sketch Decoder, provides a human-understandable window into model “thought,” which is valuable for both debugging and user-facing applications. Figures such as Figure 4 (qualitative visual examples) and Figure 7 (quantitative/qualitative validation of Sketch Decoder) showcase these strengths.

**Weaknesses:**

1. This work may lack innovation. While the integration of a visual sketchpad is interesting, a number of recent efforts (e.g., Visual Sketchpad (Hu et al., 2024), Interactive Sketchpad (Chen et al., 2025), Visual-ARFT (Liu et al., 2025)) are closely aligned. The paper’s distinction lies more in implementation detail than breakthrough conceptual separation, and this is not sufficiently articulated in the text. The positioning versus directly relevant prior work is still somewhat superficial; e.g., Section 5 makes mention of “sketchpad” approaches, but does not substantively contrast with these closest competitors in either methods or results. The core innovation is incremental rather than transformative.
2. All empirical evidence is centered on the synthetically generated MAZEPLANNING dataset. While this allows controlled analysis, the scope is narrow—visual chain-of-thought for maze navigation is a particular reasoning task, and the broader applicability (e.g., VQA, real-world visual planning, or naturalistic tasks) remains unproven. This undermines the paper’s claims of “broad applicability.”
3. The quantitative improvements from Latent Sketchpad are modest. For instance, Table 1 and Table 3 report absolute gains that are typically <3% in success or progress rate when augmenting baselines. For Qwen2.5-VL in Table 1 the gains are even smaller (less than half a percent in most settings), which is unlikely to move the needle in practical terms. The interpretability benefits are not clearly benchmarked against user or task-driven utility.
4. While the paper is generally carefully written, some aspects of the latent regression formulation and training are underspecified. For instance, Equation for $\mathcal{L}_{reg}$ on Page 4 is kept generic (“various similarity or distance measures”), but in ablations it is revealed L1 is best—this could be systematized. The causal attention mechanisms, as shown in Figure 2, are described at a high level but without clear formal specification (e.g., masking details, variable dependencies). The mapping from Vision Head latent space through the AlignerNet to VAE latent codes is abstracted, yet non-trivial for reproducibility or for adapting to different base models.
5. While Table 4 provides some insight into ablation by showing effects of connector adaptation and loss, the influence of the pretrained Sketch Decoder on reasoning performance, compared to using simpler visualizer modules, is not deeply examined. Also, how much the interpretability/visualization adds to actual reasoning success or end-user value is not directly assessed.

**Questions:**

1. What fundamental advances does Latent Sketchpad offer over Visual Sketchpad (Hu et al., 2024), Interactive Sketchpad (Chen et al., 2025), and Visual-ARFT (Liu et al., 2025)? Please detail the algorithmic or empirical distinctions—ideally supported by direct head-to-head comparison or ablation against these methods.
2. Can the authors clarify the specific masking and sequencing logic for the cross/self-attention in Vision Head (Figure 2)? Are the attention masking strategies and context construction robust to variable sequence lengths and complex, multi-turn tasks?
3. What are the technical or empirical bottlenecks preventing Latent Sketchpad from yielding larger gains on the Qwen2.5-VL backbone, especially on OOD test sets as per Table 3 and Figure 5?
4. Are there any plans to test Latent Sketchpad on broader real-world datasets or tasks (e.g., VQA, robotic planning, dialog-based spatial reasoning) to substantiate claims of “broad applicability”? What, if any, are the barriers to such generalization?
5. How sensitive are the results to different choices of the latent regression loss (e.g., cosine similarity, MSE)? Is the Vision Head prone to collapse or instability depending on training details?
6. Can the authors provide clearer insight into the limits of using SSIM for structural assessment (Figure 3), since this metric may not correlate with reasoning-relevant spatial fidelity?

---

> ### Author Response · Authors · 2025-11-20
> **Response to Reviewers [1/3]**
>
> ### [W1&Q1] Novelty and Positioning:
>
> As clarified in our *General Response (Clarification on Core Contribution, Feasibility, and Vision)*, our framework **differs fundamentally in goal, mechanism, and scope** from Visual Sketchpad and other work you have mentioned.
>
>
> **1) Conceptual distinction (what the model is doing)**
>
> - **Native visual thought vs. external visualization/tools.** Prior “sketchpad” systems expose visual traces—typically via external drawing/rasterization modules or tool APIs—to illustrate reasoning. In contrast, Latent Sketchpad equips the VLM with an **internal**, architecture-agnostic vision head which is designed to generate visual l atents natively inside the model’s autoregressive loop and used to guide reasoning, not merely to display it.
>
> - **Repurposing pretrained semantic features for generative thought.** Rather than relying on VAE-style pixel latents or external drawing tools, we **repurpose pretrained semantic features from frontier VLMs** to form **reasoning-oriented latents**. This is the core idea emphasized in our General Response and aligns our work with representation-centric directions rather than tool-use pipelines.
>
> **2) Algorithmic distinction (how it is implemented)**
>
> - **Native autoregressive visual latent generation:** We introduce a Context-Aware Vision Head that autoregressively predicts visual latents conditioned on the evolving text state, interleaving visual and textual tokens within the same decoding stream. No external renderer/tool is required during reasoning.
>
> - **Architecture-agnostic, decoder-native integration:** Because the generated visual latents live in a uniform semantic space, the same mechanism plugs into different frontier MLLMs (e.g., Qwen2.5-VL, Gemma3) without changing their backbone architecture and without external generators, satisfying the “no compromise to general reasoning” design goal stated in the Introduction and General Response.
>
> - **Interpretable rendering as a by-product.** We proposed a pretrained Sketch Decoder which maps visual latents to human-interpretable sketches for visualization only. Crucially, this decoder is not a required tool during reasoning; it is a probe for interpretability, validating that the internal latents carry structured visual content.
>
> In summary, the Latent Sketchpad is not a reimplementation of prior tool-assisted reasoning systems, but a fundamentally different architectural concept that embeds visual thought generation natively within the decoding dynamics of pretrained VLMs. By transforming pretrained semantic representations into autoregressively generated reasoning latents, our framework redefines how multimodal models can internalize visual reasoning without auxiliary renderers or tool dependencies. This distinction marks a clear departure from previous “sketchpad” paradigms. We hope this clarification resolves the reviewer’s concern regarding the novelty and conceptual contribution of our work.
>
> ---
>
> ### [W2] Broad Applicability:
>
> We would like to clarify a potential misunderstanding regarding the term “broad applicability.”
>
> Our claim does **not** refer to applicability across diverse tasks such as VQA or naturalistic planning. As clearly stated in both the **Introduction** and Section 3.2**, our claim is that Latent Sketchpad exhibits **broad applicability across different MLLMs**. And this claim is fully supported by our experimental results across distinct model families including Qwen2.5-VL and Gemma3.
>
> Regarding the focus on the MazePlanning dataset, this was a deliberate methodological choice designed to validate the core mechanism of **native visual thought generation** under controlled and challenging conditions:
>
> - A Challenging Frontier: MazePlanning represents a demanding multimodal reasoning setting that remains difficult even for frontier MLLMs equipped with tool-use abilities (e.g., o3-pro). Demonstrating gains in such a task provides strong evidence of the framework’s feasibility.
>
>
> - Data Controllability: The task offers precise control over intermediate reasoning and visual supervision, allowing us to isolate and directly evaluate the contribution of the proposed latent sketchpad mechanism.
>
>
> As highlighted in our **General Response (Clarification on Core Contribution, Feasibility, and Vision)**, this focused setup serves as a necessary foundation to establish the architectural viability of integrating native visual reasoning into existing VLMs. We view extensions to more open-ended and real-world visual reasoning tasks as an important next step for our future work, building upon this validated foundation. We hope this clarification resolves the reviewer’s concern regarding the intended scope and interpretation of “broad applicability.”

---

> ### Author Response · Authors · 2025-11-20
> **Response to Reviewers [2/3]**
>
> ### [W3] Performance Improvements:
>
>  We agree that the observed improvements may appear modest at first glance. However, this is primarily because, after fine-tuning, Gemma3 and Qwen2.5-VL already achieve very high progress rates (above 80%) on MazePlanning. At such a performance level, introducing visual thought generation mainly serves to **push the upper bound of reasoning capability**, where numerical gains tend to appear less pronounced.
>
> To more directly validate the reasoning enhancement enabled by visual thought, we have added an additional experiment in the revised version. Specifically, we trained only our Vision Head on a frozen Qwen2.5-VL to obtain the Latent Sketchpad, and then integrated it with GPT-4o for visual thought generation. The results, as presented below, show clear improvements even for this proprietary model: **Success Rate + 3.8%** and **Progress Rate + 9.06%**.
>
> | Model     | Success Rate | Progress Rate |
> |:---------:|:------------:|:-------------:|
> |  o1       | 15.20        | 35.61         |
> | GPT-4o    | 8.60         | 30.71         |
> | **GPT-4o+LS** | **12.40**        | **39.77**         |
>
> We have included the results in our updated manuscript. We hope this additional experiment could address the reviewer’s concern.
>
> ---
>
> ### [W4,Q2&Q5] Architecture Detail:
>
> We appreciate the reviewer’s detailed technical feedback and the opportunity to clarify the latent regression formulation and architectural details.
>
> **(1) Latent regression loss:** The generic phrasing in Equation (1) was intentional to emphasize the **flexibility** of our latent regression formulation, which supports various similarity or distance measures (e.g., L1, cosine similarity, MSE). In our ablation experiments, we observed that L1 performs slightly better than cosine similarity for spatial layout. However, both objectives remained stable throughout training, and we did not observe any collapse or instability in either case.
>
> **(2) Causal attention masking:** As described in Section 2.2, each token in the local context attends only to preceding tokens across the entire multimodal sequence. This design follows the **Perceiver AR [1]** architecture, which is purpose-built for **long-context autoregressive modeling**. Consequently, our causal masking strategy and context construction are naturally robust to variable sequence lengths and scalable to complex, multi-turn reasoning tasks.
>
> **(3) AlignerNet:** The AlignerNet component serves to bridge the pretrained semantic features and pretrained VAE from SDXL. As noted in Section 2.3, this implementation follows **Kosmos-G [2]**. For reproducibility, we have provided detailed code in the supplementary materials. Regarding the adaptability across base models, we have emphasized that AlignerNet generalizes across different pretrained vision encoders in our original manuscript. As demonstrated in Section 4.1 and Figure 3, our framework supports diverse architectures such as **OpenCLIP, Qwen2.5-VL, and Gemma3**, all **exhibiting consistent generalization**.
>
> [1] Hawthorne, Curtis, et al. "General-purpose, long-context autoregressive modeling with perceiver ar." International Conference on Machine Learning. PMLR, 2022.
>
> [2] Pan, Xichen, et al. "Kosmos-G: Generating Images in Context with Multimodal Large Language Models." The Twelfth International Conference on Learning Representations.
>
> ---
>
> ### [W5&Q6] Sketch Decoder:
>
> As emphasized throughout the paper (Introduction, Section 2.3, and Section 4), the Sketch Decoder does **not participate in the reasoning process**. It serves solely as an interpretability probe that translates the internally generated visual latents into human-interpretable sketches. During inference, the MLLM performs **native reasoning entirely through its autoregressive latent generation**, without any dependency on the Sketch Decoder or external rendering.
>
> Since the decoder itself is pretrained independently, we report SSIM in Figure 3 only to quantify this **reconstruction generalization**, not as an indicator of reasoning ability. The high SSIM values on unseen samples demonstrate that the decoder preserves structural fidelity across latent spaces, ensuring reliable visualization. We emphasize that SSIM serves solely as a diagnostic indicator of reconstruction fidelity, entirely independent of the reasoning process.

---

> > ### Author Response · Authors · 2025-11-20
> > **Response to Reviewers [3/3]**
> >
> > ### [Q3] Empirical Bottlenecks on Qwen2.5-VL:
> >
> > During fine-tuning, we observed a consistent numerical discrepancy between Qwen2.5-VL and Gemma3 in their visual embedding magnitudes. When the regression loss had converged, the regression loss of Qwen2.5-VL was consistently about one order of magnitude smaller than Gemma3. We hypothesize that this low-magnitude latent regime makes Qwen2.5-VL more prone to overfitting during latent regression, thereby limiting its ability to generalize to OOD samples and to effectively leverage the generated visual thought for reasoning enhancement.
> >
> > ---
> >
> > We noticed that your initial score was 6, but it was later changed to 4 before our rebuttal was available. We regret that this adjustment occurred prior to our clarification, as some of your comments suggest possible misunderstandings of our work. We hope that our detailed responses help address these points and allow for a fairer reassessment of our contribution.

---

### Official Review · Reviewer_4jS7 · 2025-10-26

**Soundness:** 1
**Presentation:** 2
**Contribution:** 2
**Rating:** 2
**Confidence:** 3

**Summary:**

This paper introduces Latent Sketchpad, a framework that equips Multimodal Large Language Models with an internal visual reasoning process inspired by human sketching. The approach integrates a Context-Aware Vision Head to generate visual latents during autoregressive reasoning and a Sketch Decoder to render these latents into interpretable sketches. The idea is creative and relevant to advancing interpretable multimodal reasoning, though several technical and experimental aspects could benefit from clarification and broader validation.

**Strengths:**

The paper addresses a timely and meaningful problem: enhancing MLLMs with visual imagination and interpretable visual reasoning, similar to human mental sketching.

The modular architecture, i.e., the Context-Aware Vision Head and Sketch Decoder, is potentially applicable across various MLLMs (e.g., Gemma3, Qwen2.5-VL).

Keeping the visual reasoning within the latent space, rather than decoding full images during reasoning can balances interpretability and computational efficiency.

**Weaknesses:**

1. Ambiguity in the description of visual latent training (Section 2.2). The explanation of the Auto-regressive Visual Latent Generation and its associated loss is somewhat confusing. It is unclear how the “target latent obtained from pretrained visual features of the vision encoder” is defined. The paper should clarify whether these visual features come from the initial input image or from intermediate reasoning steps. If the latter, the authors should consider adding a brief preliminary section what the inputs and outputs are in this training setup, to help readers grasp the data flow more intuitively.


2. Limited scope of experiments and evaluation.

    2.1 The experiments are conducted only on the MAZEPLANNING dataset. While this dataset is useful, it alone is insufficient to demonstrate generalization. It would strengthen the paper to include results on other visual reasoning tasks such as Sokoban or Sudoku.

    2.2 Additionally, the paper should explore how a model trained with visual latent reasoning performs on more general multimodal reasoning benchmarks, such as MathVista or MMMU. A discussion about potential transferability to these tasks would make the contribution more convincing.

3. Limited performance improvement on specific backbones. According to Table 1, the gain on Qwen2.5-VL is marginal (less than 0.5). This small improvement weakens the claim that the method consistently enhances reasoning ability.

**Questions:**

Please see weaknesses.

---

> ### Author Response · Authors · 2025-11-20
> **Response to Reviewers**
>
> ### [W1] Training Details:
>
> We thank the reviewer for this valuable comment and apologize for the ambiguity in our original description. As illustrated in Figure 2, the target latent indeed comes from the intermediate visual reasoning steps, rather than the initial input image. We have revised Section 2.2 to clarify this data flow and explicitly describe how the target latents are obtained and supervised. We sincerely thank the reviewer for this helpful suggestion, which has improved the clarity of our presentation.
>
> ---
>
> ### [W2]: 2.1  Scope of Experiments
>
> The current focus on MazePlanning was a deliberate methodological choice essential for validating the feasibility of **native visual thought generation**. MazePlanning provides a domain that is both **challenging**—even for frontier MLLMs with tool-use abilities (e.g., o3-pro)—and experimentally **controllable**, enabling precise supervision of intermediate reasoning and visualization. This controlled setup allows us to isolate the contribution of the proposed Latent Sketchpad and establish architectural viability before extending to broader domains.
>
> The reviewer’s suggested tasks (e.g., Sokoban, Sudoku) indeed represent interesting next steps. However, we lack multimodal trajectory training data of these tasks, since our framework relies on pretrained MLLMs while the **vision head is trained from scratch**. We plan to explore these tasks in future work.
>
> ---
>
> ### [W2]: 2.2 Discussion on Potential Transferability
>
> Regarding general multimodal reasoning benchmarks such as MathVista and MMMU, as stated in our Introduction and General Response, our framework is designed to **enable frontier VLMs with visual thought generation without compromising their general reasoning capability**. For simple tasks like standard perception VQA, explicit visual thought generation is often unnecessary. Ideally, as pretrained backbones continue to improve, models will perform perfect on simple tasks and integrate with our Latent Sketchpad when multimodal thinking is required. We thank the reviewer for the suggestion and have added a discussion on this potential transferability in Appendix C.7 of updated manuscript.
>
> ---
>
> ### [W3] Performance Improvement:
>
> We agree that the improvement on Qwen2.5-VL appears modest in Table 1 (less than 0.5). However, this is largely due to the fact that, after fine-tuning, Qwen2.5-VL already achieves a very high progress rate (over 80%) on the MazePlanning benchmark. At such a strong baseline, introducing visual thought generation primarily serves to **push the upper bound of the model’s reasoning capacity**, where the remaining performance margin is inherently narrow and thus less reflected by numerical gain.
> To more directly verify the reasoning enhancement brought by our method, we conducted an additional experiment in the revised version. We trained only our Vision Head on a frozen Qwen2.5-VL to obtain the Latent Sketchpad, and then integrated it with GPT-4o for visual thought generation. As reported in the updated Table 1, this setup yields clear improvements even for a proprietary model: **Success Rate +3.8% and Progress Rate +9.06%**. These results (also presented below) further support our contribution.
>
> | Model     | Success Rate | Progress Rate |
> |:---------:|:------------:|:-------------:|
> |  o1       | 15.20        | 35.61         |
> | GPT-4o    | 8.60         | 30.71         |
> | **GPT-4o+LS** | **12.40**        | **39.77**         |
>
> We hope this additional experiment could address the reviewer’s concern.

---

### Official Review · Reviewer_FKhg · 2025-11-01

**Soundness:** 2
**Presentation:** 3
**Contribution:** 2
**Rating:** 2
**Confidence:** 4

**Summary:**

This paper introduces latent Sketchpad, specifically on visual maze problems. They introduce two components: a Context-aware Vision Head that produces visual representations, and a pretrained Sketch Decoder that renders these into human-interpretable images. They create a new dataset MazePlanning and evaluate their method on this new dataset.

**Strengths:**

1. provides a method to conduct latent sketchpad by designing a Context-aware Vision Head, and a Pretrained Sketch Decoder.
2. Provides a 47.8K training data for maze puzzle solving.
3. Experiments upon Gemma3-12B and Qwen2.5-VL-7B, shows improvement (+0.39-+2.2%) on the proposed dataset.

**Weaknesses:**

The biggest weakness is that the experiment is not clear and some citations are missing.
1. For the experiments, the evaluations are done on generated text action sequences (Sec 3.1). For the result in Table 1, it shows text-only output and interleaved text-image output. What does it mean? Does it mean still only the text is evaluated the interleaved output setting? Also, it is unclear what training data (text only data v.s. multimodal data) is used for training for these two outputs. Do you use text-only data for text-only output?
2. While the settings are unclear, it seems most improvement hover around 0.5%. It doesn't seem significant. Does this result consider generate multiple outputs and than evaluate? If not, what would be the new scores after generating multiple outputs than eval?
3. Baseline missing. It seems unified model should be added to the baselines e.g. MetaMorph, Bagel, Janus Pro. Also Gemini 2.5 Pro and GPT 5. Also, should include [Imagine while Reasoning in Space: Multimodal Visualization-of-Thought] as a baseline because you're solving same tasks.
4. Evaluation Benchmark too few, only evaluating on your proposed benchmark cannot prove the efficacy of the method. Should consider also evaluate on other benchmarks, e.g. [VISUALPUZZLES: Decoupling Multimodal Reasoning Evaluation from Domain Knowledge], [VGRP-Bench: Visual Grid Reasoning Puzzle Benchmark for Large Vision-Language Models]... etc.
5. Missing citations. Especially, (a) should be cited because that's usually where "sketchpad" comes from.  a. [Visual Sketchpad: Sketching as a Visual Chain of Thought for Multimodal Language Models] from NeurIPS 2024. b. [Perception Tokens Enhance Visual Reasoning in Multimodal Language Models] from CVPR 2025. c. [Machine Mental Imagery: Empower Multimodal Reasoning with Latent Visual Tokens] d. [VISUALPUZZLES: Decoupling Multimodal Reasoning Evaluation from Domain Knowledge]

**Questions:**

Please see weakness. Happy to raise score if these are addressed.

---

> ### Author Response · Authors · 2025-11-20
> **Response to Reviewer [1/2]**
>
> ### [W1] Evaluation Details:
>
> We thank the reviewer for pointing out this concern and appreciate the opportunity to clarify. As stated in **Section 3.1 (Evaluation Metrics)**, the Success Rate and Progress Rate reported in Table 1 are computed solely from the final textual outputs. These metrics evaluate whether the model’s generated textual action sequences correctly solve the task. In contrast, the generated visual thoughts are quantitatively analyzed in **Section 4.2**, where we report **Layout Consistency Rate** and **Visual Success Rate** in Table 2 to assess the quality and coherence of the visual outputs.
>
> Regarding the training data, as described in **Section 3.1** and further detailed in **Appendix B.1** (third paragraph), we adopt a unified fine-tuning scheme that equips a single model to operate in both text-only and interleaved modes. Specifically, the backbone is trained on a balanced mixture of 50% text-only data and 50% interleaved text–image data. During inference, the model can flexibly switch modes:
>
> - For text-only output, it operates identically to standard decoding.
> - For interleaved multimodal CoT, a special token \<start_of_image\> is automatically inserted during generation.
>
> All implementation details are provided in Appendix B, and we have made these explanations more explicit in the updated manuscript. We hope this clarification resolves the reviewer’s concerns.
>
> ---
>
> ### [W2] Experimental Settings:
>
> We appreciate the reviewer’s comment and agree that the reported improvements may appear modest at first glance. However, this is primarily because, after fine-tuning, Gemma3 and Qwen2.5-VL already achieve very high progress rates (above 80%) on MazePlanning. At such a high baseline, introducing visual thought generation mainly serves to **push the upper bound of reasoning capability**, where numerical gains tend to appear less dramatic.
>
> To further substantiate the reasoning enhancement enabled by the proposed mechanism, we have added an additional experiment in the revised version. Specifically, we trained only our Vision Head on a frozen Qwen2.5-VL to obtain the Latent Sketchpad, and integrated it with GPT-4o for visual thought generation. As shown in the updated Table 1, this setup yields clear gains even for a proprietary model: **Success Rate + 3.8%** and **Progress Rate + 9.06%**.
> Regarding the evaluation setting, we use **greedy decoding** throughout all experiments and do not generate multiple outputs per query. We intentionally adopt this setting because MazePlanning tasks are deterministic and goal-oriented, which does not need creative generation.
> We hope this clarification resolves the reviewer’s concern.
>
> ---
>
> ### [W3] Baseline:
>
> We thank the reviewer for this valuable suggestion regarding the choice of baselines. However, the unified models mentioned (e.g., MetaMorph, Bagel, Janus Pro) do **not support native interleaved generation**—that is, they cannot produce textual and visual reasoning steps within a single autoregressive decoding process. MetaMorph fails to converge on the maze planning task, as it does not take the image context into account during training, leading to unstable optimization. In contrast, Bagel and Janus Pro rely on additional VAE-based visual features and thus treat text generation and image generation as two separate tasks rather than a unified autoregressive process. Moreover, the official implementations of these models do not provide inference code for interleaved generation, further confirming that they are not designed to handle such mixed-modality reasoning in practice. Since our framework specifically targets native visual thought generation integrated inside existing VLMs as detailed in our General Response, these models are not directly comparable in scope or mechanism.
>
> The cited work *Imagine while Reasoning in Space: Multimodal Visualization-of-Thought* (MVoT) also differs substantially in nature. MVoT introduces a new **reasoning method** rather than a trainable model and, importantly, the authors have **not released any model weights and their dataset**. Additionally, we have **already included a unified MLLM baseline (Liquid)** following the setting of MVoT in our original manuscript (Appendix C.1), with corresponding results presented in Tables 11 and 12. The results demonstrate that built on frontier VLMs like Gemma3, our Latent Sketchpad achieves better performance and generalization.
>
> Regarding Gemini 2.5 Pro and GPT-5, we note that our experiments on **GPT-4o, o1, o4-mini, and o3-pro** already illustrate that proprietary models struggle with complex and dynamic multimodal reasoning tasks such as MazePlanning. Even though Gemini 2.5 Pro and GPT-5 may internally invoke image generation modules to produce visual outputs, we found that the generated images perform poorly on the MazePlanning task, failing to demonstrate coherent spatial reasoning or consistent path planning capabilities.

---

> > ### Author Response · Authors · 2025-11-20
> > **Response to Reviewers [2/2]**
> >
> > ### [W4] Evaluation Benchmark:
> >
> > The current focus on MazePlanning is a deliberate methodological choice essential for validating the core mechanism of **native visual thought generation**. To this end, we selected a domain that is both sufficiently challenging and experimentally controllable:
> >
> > - A Challenging Frontier: MazePlanning represents a demanding multimodal reasoning setting that remains difficult even for frontier MLLMs equipped with tool-use abilities (e.g., o3-pro). Demonstrating gains in such a task provides strong evidence of the framework’s feasibility.
> >
> >
> > - Data Controllability: The task offers precise control over intermediate reasoning and visual supervision, allowing us to isolate and directly evaluate the contribution of the proposed latent sketchpad mechanism.
> >
> >
> > As highlighted in our **General Response (Clarification on Core Contribution, Feasibility, and Vision)**, this focused setup was essential to establish architectural viability before extending to broader, open-ended reasoning domains.
> > We also note that the benchmarks suggested by the reviewer (e.g., VISUALPUZZLES, VGRP-Bench) require task-specific visual thought representations and corresponding dataset construction to train the generation module. Since our framework is built upon pretrained MLLMs while the **vision head is trained from scratch**, developing and aligning such datasets lies beyond the current scope. Nonetheless, we fully agree that exploring these additional benchmarks is a valuable next step, and we plan to extend our framework to these domains in future work.
> >
> > ---
> >
> > ### [W5] Missing Citations:
> >
> > We thank the reviewer for pointing out these relevant works and for the helpful citation suggestions. We would like to clarify that the concept of a “sketchpad” in our paper does **not** originate from recent multimodal reasoning studies, but rather from **cognitive psychology**. As stated in our Introduction, the term is inspired by the visuospatial sketchpad in the multicomponent model of working memory proposed by Raymond Bruyer and Jean-Christophe Scailquin, “The visuospatial sketchpad for mental images: Testing the multicomponent model of working memory,” Acta Psychologica, 98(1):17–36, 1998. Additionally, we use the term "sketchpad" purely for readability and intuitive understanding, **rather than to introduce a new concept**.
> >
> > Nonetheless, we greatly appreciate the reviewer’s recommendation and have updated the Related Work section to include these work.

---

### Official Review · Reviewer_PDJW · 2025-11-02

**Soundness:** 3
**Presentation:** 3
**Contribution:** 2
**Rating:** 4
**Confidence:** 3

**Summary:**

This paper proposes a framework that enables Multimodal Large Language Models (MLLMs) to “think visually” during reasoning. Inspired by human mental sketching, the method adds a Context-Aware Vision Head that generates visual latents interleaved with textual reasoning, and a pretrained Sketch Decoder that translates these latents into interpretable sketches. The authors introduce a new MAZEPLANNING dataset designed to test multimodal reasoning involving both text and spatial planning. Experiments on models such as Gemma3 and Qwen2.5-VL show that integrating the Latent Sketchpad slightly improves reasoning accuracy while producing interpretable visual traces. Overall, the work contributes a plug-and-play method to integrate visual imagination into MLLMs without retraining the backbone.

**Strengths:**

1. The paper’s originality lies in its attempt to simulate a “visual thinking” process within MLLMs by introducing a latent sketching mechanism—a creative and human-inspired idea that connects internal representation learning with interpretable visual reasoning.
2. The quality of the work is solid, with a clear architectural design combining a context-aware vision head and a pretrained sketch decoder, along with empirical validation on the proposed MazePlanning benchmark and existing reasoning tasks.
3. In terms of clarity, the paper is well-structured and communicates its motivation and framework intuitively, aided by clear figures illustrating how sketches emerge during reasoning.
4. Regarding significance, the work offers a promising step toward more interpretable multimodal reasoning and introduces a framework that can be easily adapted across models without retraining, making it an interesting and practical contribution to the MLLM research community.

**Weaknesses:**

1. While the idea of a “latent sketchpad” is creative, the paper’s novelty is somewhat limited, as related works such as Visual Chain-of-Thought (Zhou et al., 2023), MM-ReAct (Yao et al., 2023), and Sketch-Guided CoT (Luo et al., 2024) have also explored visual reasoning traces or intermediate visualizations.
2. The experiments are relatively narrow—focused mainly on MazePlanning and a few reasoning benchmarks, the OOD performance also degrades significantly. Maybe the type of dataset can be extended to broader domains.
3. The observed performance gains are modest, suggesting that the sketching component currently serves more as a visualization tool than a strong reasoning enhancement.

**Questions:**

NA

---

> ### Author Response · Authors · 2025-11-20
> **Response to Reviewers**
>
> ### [W1] Concern about Novelty:
>
> We sincerely thank the reviewer for the positive recognition that *“the idea of a latent sketchpad is creative.”* We believe, however, that there may be some misunderstanding regarding the conceptual scope and novelty of our framework.
>
> As clarified in both our **introduction (lines 77–78)** and the **General Response (Clarification on Core Contribution, Feasibility, and Vision)**, our goal is to **repurpose pretrained semantic features from frontier vision–language models (VLMs)** to enable **native visual thought generation**. This differs fundamentally from prior works such as Visual Chain-of-Thought, MM-ReAct, and Sketch-Guided CoT, which focus on externally visualizing reasoning traces or tool-based sketch generation.
>
> In contrast, our framework equips existing VLMs (e.g. Qwen2.5-VL, Gemma3) with an **internal, architecture-agnostic visual reasoning mechanism** that integrates latent visual generation directly into the autoregressive reasoning loop. This design expands the functional role of visual representations beyond perception toward *generative internal reasoning*, a capability not explored in prior approaches.
>
> We appreciate the reviewer’s valuable pointers and have updated the Related Work section to explicitly discuss and contrast our framework with these studies.
>
> ---
>
> ### [W2]  Experiments Scope:
>
>  We appreciate the reviewer’s comment regarding the experimental scope. The current focus on MazePlanning is a deliberate methodological choice essential for validating the core mechanism of **native visual thought generation**. To this end, we selected a domain that is both sufficiently challenging and experimentally controllable:
>
> - **A Challenging Frontier:** MazePlanning represents a demanding multimodal reasoning setting that remains difficult even for frontier MLLMs equipped with tool-use abilities (e.g., o3-pro). Demonstrating gains in such a task provides strong evidence of the framework’s feasibility.
> - **Data Controllability:** The task offers precise control over intermediate reasoning and visual supervision, allowing us to isolate and directly evaluate the contribution of the proposed latent sketchpad mechanism.
>
> As highlighted in our General Response(Clarification on Core Contribution, Feasibility, and Vision), this focused setup was essential to establish architectural viability before expanding to broader, open-ended reasoning domains. We view this work as laying the conceptual and empirical foundation for such extensions, which we consider a critical direction for future research. We hope the reviewer could consider our work in this context, and we will extend our work to a wider range of reasoning tasks in future research.
>
> ---
>
> ### [W3] Performance Gain:
>
> We agree that the observed improvements may appear modest at first glance. However, this is primarily because, after fine-tuning, Gemma3 and Qwen2.5-VL already achieve very high progress rates (above 80%) on MazePlanning. At such a performance level, introducing visual thought generation mainly serves to **push the upper bound of reasoning capability**, where numerical gains tend to appear less pronounced.
>
> To more directly validate the reasoning enhancement enabled by visual thought, we have added an additional experiment in the revised version. Specifically, we trained only our Vision Head on a frozen Qwen2.5-VL to obtain the Latent Sketchpad, and then integrated it with GPT-4o for visual thought generation. The results, as presented below, show clear improvements: **Success Rate + 3.8%** and **Progress Rate + 9.06%**. With Latent Sketchpad, GPT-4o achieves performance comparable to reasoning models and even has surpassed o1.
>
> | Model     | Success Rate | Progress Rate |
> |:---------:|:------------:|:-------------:|
> |  o1       | 15.20        | 35.61         |
> | GPT-4o    | 8.60         | 30.71         |
> | **GPT-4o+LS** | **12.40**        | **39.77**         |
>
> We have included the results in our updated manuscript. We hope this additional experiment could address the reviewer’s concern.

---

### Author Response · Authors · 2025-11-20

## Clarification on Core Contribution, Feasibility, and Vision

Our primary contribution lies in **repurposing pretrained semantic features from frontier vision-language models (VLMs)** to enable *native visual thought generation*, using sketches as a tractable and interpretable proxy.


A key advantage of this approach is its reliance on **uniform visual representations**—unlike methods that depend on VAE-based generative features, our framework is **architecture-agnostic**, making it more flexible and compatible with diverse model families.

Conceptually, our approach parallels the recent direction introduced by **Representation Autoencoders (RAEs)** [1].


RAE replaces the traditional VAE used for image generation with pretrained representation encoders such as **DINO** or **SigLIP**, achieving semantically meaningful latent spaces for diffusion transformers.


Similarly, our framework leverages pretrained semantic features to **enable reasoning-oriented latent representations** within existing VLMs.

A significant part of our contribution is a **comprehensive examination of architectural feasibility**. We extensively validated this approach across:

* **Diverse VLMs:** (e.g., Qwen2.5-VL, Gemma3), by integrating a vision head and training the model to support native visual thought during decoding—originally a text-only process.


* **Various Semantic Feature Sets:** (e.g., OpenCLIP, SigLIP), providing a series of pretrained sketch decoders that demonstrates strong generalization and robust reconstruction ability.

This foundational work demonstrates the **architectural viability** of our method and, we believe, **paves the way for future exploration** by the community. We position this direction alongside promising approaches like RAE and emphasize the potential of semantic features for **unified visual understanding and generative reasoning**.

---

## Distinction from Interleaved Reasoning and Tool-Use

We emphasize that our main contribution **is not a new interleaved reasoning paradigm** or a novel tool-use VLM.

Instead, our framework **enables advanced reasoning capabilities natively within existing VLMs**.


The critical distinction is that our method fosters internal visual reasoning **without relying on external, pixel-level image generation modules or auxiliary tools**.

References:


[1] Zheng, Boyang, et al. "Diffusion Transformers with Representation Autoencoders." arXiv preprint arXiv:2510.11690 (2025).

---

### Meta-Review · Area_Chair_2Cbj · 2026-01-03

**Summary:**

The reviewers' primary concerns are around three main areas: the novelty of the contribution, the narrow experimental scope, and the modest performance gains.
- *Novelty*: PDJW, FKhg, QkdV felt the work was incremental, citing a number of recent papers with "sketchpad" in their titles or related concepts (e.g., Visual Chain-of-Thought, MM-ReAct). They argued that the paper did not sufficiently articulate its conceptual separation from these prior works, which also explore intermediate visual reasoning traces.
- *Experimental scope*: A consensus concern across all reviewers (PDJW, FKhg, 4jS7, QkdV) was the evaluation's reliance on a single proposed, and synthetic dataset (MazePlanning). They argued this was insufficient to demonstrate the generalization of the proposed framework. Evaluation on other reasoning benchmarks like VISUALPUZZLES, Sokoban, or even general MLLM benchmarks like MMMU is necessary.
- *Performance gains*: The reviewers (PDJW, FKhg, 4jS7, QkdV) consistently pointed out that the performance improvements on the backbone models (Gemma3, Qwen2.5-VL) were marginal. This led them to question whether the Latent Sketchpad provides a significant reasoning enhancement or functions primarily as an interpretability tool.

**Reviewer Concerns:**

Addressed Concerns:
- *Novelty*: This was the most critical concern, and the authors addressed it by clarigying that their core contribution is not merely generating visual traces, but enabling native visual thought generation within the latent space of an MLLM, repurposing pretrained semantic features. They convincingly distinguished their work from tool-use or external rendering approaches by emphasizing that their method is architecture-agnostic, integrated directly into the autoregressive decoding loop, and does not rely on external modules during reasoning.
- *Modest performance gains* (PDJW, FKhg, 4jS7, QkdV): The authors' most impactful rebuttal point was the inclusion of a new experiment. They argued the small gains on Gemma3/Qwen2.5-VL were due to the models already achieving near-SOTA performance on the task after fine-tuning. To demonstrate the module's true value, they integrated their Latent Sketchpad with GPT-4o, a model that performs poorly on MazePlanning. The results showed better improvement (Success Rate +3.8%, Progress Rate +9.06%), providing evidence that the framework enhances reasoning capabilities.
- *Experimental/Technical clarity* (FKhg, 4jS7): The authors successfully clarified the evaluation metrics, the training data composition, and the technical details of the latent target generation, resolving the confusion expressed by the reviewers.

Outstanding Concerns:
- *Narrow experimental scope* (All Reviewers): While the authors provided a reasonable justification for focusing on MazePlanning—namely, that it serves as a controlled and challenging environment for a foundational feasibility study—the concern about generalization to other tasks and real-world scenarios remains.
- *Missing baselines*: As suggested by FKhg that unified model-based methods should be added for comparisons as well. The baselines are very limited in current version.

**Reviewer Scores:**

- Reviewer PDJW (Initial: 4): The authors addressed the novelty issue but failed to address the experimental issue, which can make the score increasing pretty hard.

- Reviewer FKhg (Initial: 2): The rebuttal provides enough experimental and evaluation details but fail to provide more valid baselines and demonstrate the generalization of this method to general visual reasoning. I anticipate that the reviewer would raise the score but it is challenging to assign a positive score.

- Reviewer 4jS7 (Initial: 2): The authors clarified the technical ambiguity and provided experiments to solve the performance concerns. However, similar with the last reviewer, the marginal improvement on a SOTA model and the limited evaluation scope make the score raising difficult.

- Reviewer QkdV (Initial: 6): This reviewer was already leaning towards acceptance. The authors' rebuttal was equally detailed, particularly on the crucial point of novelty and positioning.

In general, due to marginal performance gains, a narrow evaluation scope, and insufficient baseline comparisons, the paper does not currently meet the acceptance criteria for ICLR.

---

### Decision · Program_Chairs · 2026-01-26

Reject